# Intratumoural immune heterogeneity as a hallmark of tumour evolution and progression in hepatocellular carcinoma

Phuong H. D. Nguyen[1,16], Siming Ma[2,16], Cheryl Z. J. Phua[2], Neslihan A. Kaya[2,3], Hannah L. H. Lai[2], Chun Jye Lim[1], Jia Qi Lim[2], Martin Wasser[1], Liyun Lai[1], Wai Leong Tam[2,3,4,5], Tony K. H. Lim[6,7], Wei Keat Wan[6,7], Tracy Loh[6,7], Wei Qiang Leow[6,7], Yin Huei Pang[8], Chung Yip Chan[7,9,10], Ser Yee Lee[7,9,10], Peng Chung Cheow[7,9,10], Han Chong Toh[7,9], Florent Ginhoux[1,11], Shridhar Iyer[12], Alfred W. C. Kow[12], Yock Young Dan[13], Alexander Chung[7,9,10], Brian K. P. Goh[7,9,10], Salvatore Albani[1,16], Pierce K. H. Chow[7,9,10,16✉], Weiwei Zhai[2,14,15,16✉] & Valerie Chew[1,16✉]

The clinical relevance of immune landscape intratumoural heterogeneity (immune-ITH) and its role in tumour evolution remain largely unexplored. Here, we uncover significant spatial and phenotypic immune-ITH from multiple tumour sectors and decipher its relationship with tumour evolution and disease progression in hepatocellular carcinomas (HCC). Immune-ITH is associated with tumour transcriptomic-ITH, mutational burden and distinct immune microenvironments. Tumours with low immune-ITH experience higher immunoselective pressure and escape via loss of heterozygosity in human leukocyte antigens and immunoediting. Instead, the tumours with high immune-ITH evolve to a more immunosuppressive/exhausted microenvironment. This gradient of immune pressure along with immune-ITH represents a hallmark of tumour evolution, which is closely linked to the transcriptome-immune networks contributing to disease progression and immune inactivation. Remarkably, high immune-ITH and its transcriptomic signature are predictive for worse clinical outcome in HCC patients. This in-depth investigation of ITH provides evidence on tumour-immune co-evolution along HCC progression.

[1] Translational Immunology Institute (TII), SingHealth-DukeNUS Academic Medical Centre, Singapore, Singapore. [2] Genome Institute of Singapore (GIS), Agency for Science, Technology and Research (A*STAR), Singapore, Singapore. [3] School of Biological Sciences, Nanyang Technological University Singapore, Singapore, Singapore. [4] Cancer Science Institute of Singapore, National University of Singapore, Singapore, Singapore. [5] Department of Biochemistry, Yong Loo Lin School of Medicine, National University of Singapore, Singapore, Singapore. [6] Department of Pathology, Singapore General Hospital, Singapore, Singapore. [7] Duke-NUS Medical School, Singapore, Singapore. [8] Department of Pathology, National University Hospital Singapore, Singapore, Singapore. [9] National Cancer Centre, Singapore, Singapore. [10] Department of Hepatopancreatobiliary and Transplant Surgery, Singapore General Hospital, Singapore, Singapore. [11] Singapore Immunology Network (SIgN), A*STAR, Singapore, Singapore. [12] Department of Medicine, Yong Loo Lin School of Medicine, National University Hospital Singapore, Singapore, Singapore. [13] Division of Gastroenterology and Hepatology, National University Hospital Singapore, Singapore, Singapore. [14] Key Laboratory of Zoological Systematics and Evolution, Institute of Zoology, Chinese Academy of Sciences, Beijing, China. [15] Center for Excellence in Animal Evolution and Genetics, Chinese Academy of Sciences, Kunming, China. [16] These authors contributed equally: Phuong H. D. Nguyen, Siming Ma, Salvatore Albani, Pierce K. H. Chow, Weiwei Zhai, Valerie Chew. ✉email: pierce.chow.k.h@singhealth.com.sg; zhaiww1@gis.a-star.edu.sg; valerie.chew.s.p@singhealth.com.sg

Hepatocellular carcinoma (HCC) is known to be a heterogeneous tumour derived primarily from a background of chronic liver inflammation with various etiopathogenesis including chronic viral hepatitis infection, alcoholism and fatty liver diseases[1]. Due to the heterogenous nature and hence the limited options for targeted treatment, HCC remains the third leading cause of cancer mortality globally[2]. The recent success of immunotherapy in HCC benefits only up to 20% of the patients who would respond to the anti-PD-1 immune-checkpoint blockade (ICB)[3,4]. Hence, the current immunotherapy landscape leans towards combination therapy with enhanced clinical efficacy, such as that demonstrated by the recently approved atezolizumab (anti-Programmed death-ligand 1 [PD-L1]) and bevacizumab (anti-Vascular Endothelial Growth Factor A [VEGFA]) combination therapy for advanced HCC from the phase III (IMbrave150) trial[5]. Moreover, a recent biomarker study from liver cancer patients treated with ICB demonstrated that low tumour cell transcriptomic diversity and cytolytic activity of CD8 + T cells predict their therapeutic response[6]. This warrants a deeper understanding of the complex nature of immune microenvironment and its relationship with tumour genomic profiles in a spatio-temporal manner.

The genomic intratumoural heterogeneity (ITH) was previously described as an important hallmark of tumour evolution and cancer progression[7] including in HCC[8,9]. On the flip side, the biological and clinical relevance of ITH in the tumour microenvironment (TME), based upon degree of heterogeneity in the spatial distributions and the phenotypes of tumour-infiltrating leukocytes (TILs), is not known. It was previously described that distinct histological TME could impact on clinical outcome of HCC[10,11], whereas multiomic analyses also described intensive ITH in TME of HCC[12]. Other recent studies using immunogenomics approach addressed how the immune landscape contributes to genomic ITH in ovarian cancer[13] and HCC[14]. Despite all that, the clinical impact and role of immune-ITH in tumour evolution remain unexplored. Furthermore, tumour evolution or immunoediting driven by immunoselective pressure was previously shown in various cancers[15,16]; it is however not known whether immune-ITH is linked to TME with different immunoselective pressure, which drives tumour evolution. Given the multistep nature of carcinogenesis and disease progression in HCC, it will be important to study and understand the evolution of its immune microenvironment along with tumour genomic evolution.

Our study aims to fill the knowledge gap in the field of tumour ITH and to examine the significance of immune-ITH in tumour evolution and disease progression. Using multi-sectoring and multi-omics approaches on different regions from a single HCC tumour, we found a marked degree of immune-ITH, which is correlated to tumour transcriptomic-ITH. Concurrently, the overall TME shows decreasing immunoselective pressure with increased immune-ITH, indicating an immune evolution towards immune exhaustion/suppression. Along with this differential immunoselective pressure, tumour evolve with distinct escape strategies. We also uncovered immune-ITH-related transcriptome-immune networks and the distinct molecular pathways involved in dictating the disease progression and immune status. The current findings demonstrate the remodelling of immune landscape with increased immune-ITH as another dimension in tumour-immune co-evolution, which can be harnessed as a predictive signature for tumour progression.

## Results

**Significant degree of immune-ITH in HCC**. Based on our previous discovery of significant genomic ITH and its impact on evolution trajectory in HCC[9], we aimed to examine the degree and implication of ITH in the immune landscapes from multiple regions within an HCC tumour. Following strict sampling protocol of two to five regions per tumour (Supplementary Fig. 1a, b), we prospectively collected a total of 95 tumour sectors (T) with its adjacent non-tumour liver tissues (N) and peripheral blood (P) from 28 HCC patients who underwent surgical resection as the first-line therapy without any prior treatment (Supplementary Table 1). The samples from the same region were analysed by cytometry by time-of-flight (CyTOF), whole genome sequencing (WGS) and RNA sequencing for their immunomic, genomic and transcriptomic profiles, respectively (Fig. 1a). CyTOF analysis was performed using 38 surface or intracellular immune markers (Supplementary Table 2) as previously described[17].

We employed Phenograph clustering[18] and our in-house CyTOF analytics pipeline[19] on the data generated from all tumour sectors and identified 30 immune cell clusters (Fig. 1b). The clusters were assigned to major immune lineages according to their lineage marker expressions (Fig. 1c). The varying proportions of these clusters across sectors from the same tumour showed different degree of immune-ITH (Fig. 1d and Supplementary Fig. 2a). Next, we examined the ITH by manual gating of all major TIL subsets identified as the key global representative of TME in HCC from our previous study[17] (Supplementary Fig. 3a). Again, we observed significant variations in the proportions (Fig. 1e) and the variances (Supplementary Fig. 3b) of these 15 immune subsets in the TME of HCC, indicating varying degree of immune-ITH. Next, to systematically quantify for immune-ITH, we compared the proportions of these 15 immune subsets in pairwise manner across all tumour sectors from each tumour using Spearman's correlation coefficient ($\rho$), which measures degree of association or homogeneity[20]; the immune-ITH scores (degree of heterogeneity) were reported as 1 $- \rho$ (Supplementary Fig. 4a). HCC tumours showed varying degree of immune-ITH (Fig. 1e, bottom) and the immune-ITH scores according to t-distributed stochastic neighbor embedding (tSNE) clusters or manual gating of 15 immune subsets showed good concordance (Fig. 1f). We also calculated immune-ITH scores using previously described Euclidean distance[21] and demonstrated a high correlation between the two scoring methods ($\rho = 0.99$, Fig. 1g) resulting in the same immune-ITH groupings of HCC tumours according to their respective medians (Supplementary Fig. 5a).

To validate whether ITH was also reflected by tissue immune cell density, we next examined the heterogeneity in the densities of the $CD4^+$ and $CD8^+$ T cells within tumour tissues using multiplex immunohistochemistry (mIHC) (Fig. 1h, left). We assigned ITH scoring by calculating the standard deviation (SD) of $CD4^+$ and $CD8^+$ T-cell densities from ten regions of each tumour. Indeed, we observed a significant correlation in the degrees of immune-ITH based on CyTOF (proportions of immune subsets) or tissue mIHC (cell densities) (Fig. 1h, right), both demonstrating consistent degree of Immune-ITH.

Taken together, the above data demonstrated significant immune-ITH within HCC tumours.

**Tumour evolutionary events are linked to immune-ITH**. Next, we aim to explore immune-ITH as a hallmark of tumour evolution along tumour mutational trajectory. First, we compared immune-ITH with genomic (DNA)- and transcriptomic (RNA)-ITH, which were shown to be an important hallmark of tumour evolution[7,9]. We constructed the phylogenetic trees for the RNA and DNA profiles of the tumours with low or high immune-ITH using median immune-ITH score as the cut off (Fig. 2a). The

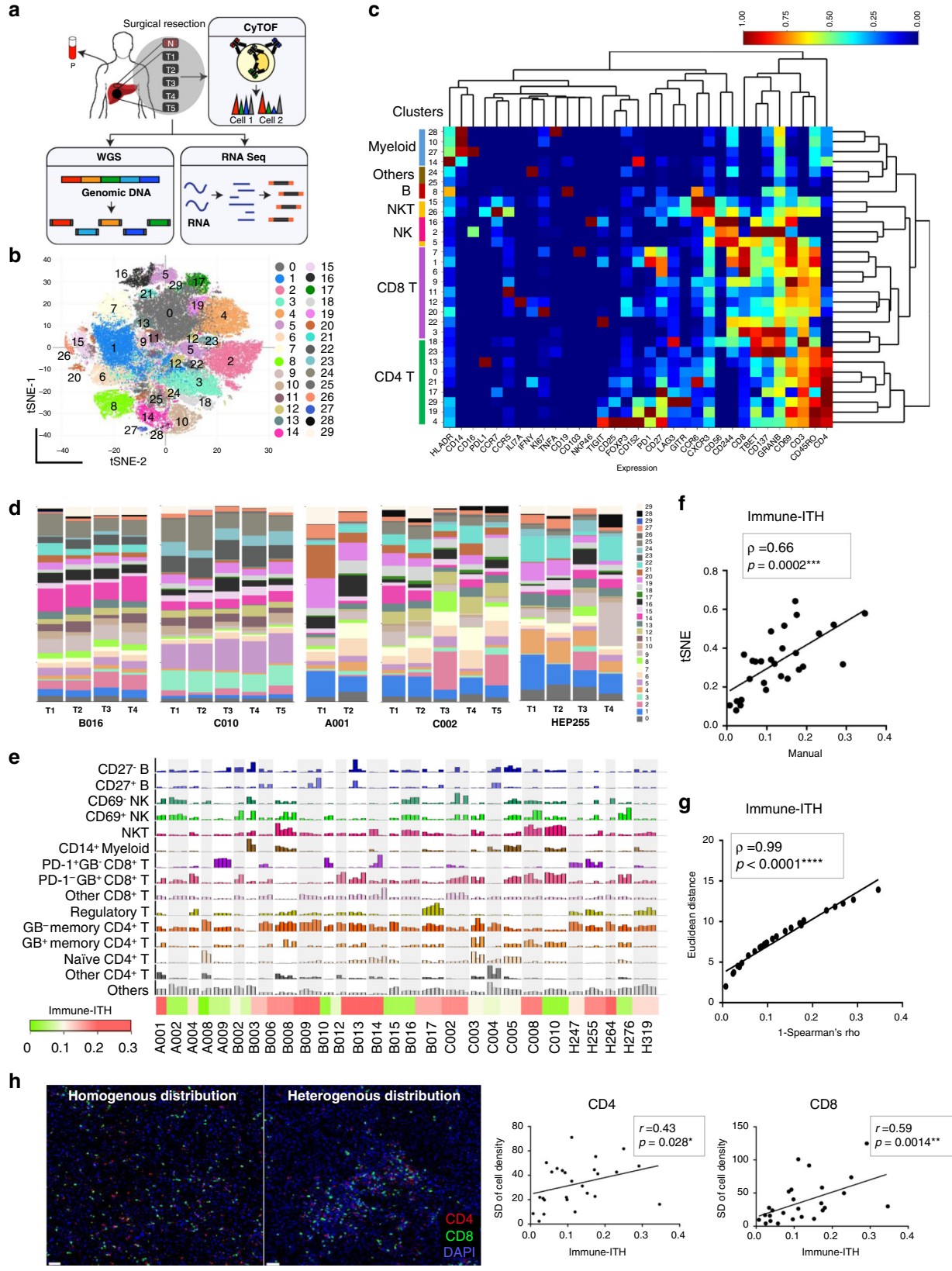

phylogenetic trees illustrated the evolutionary relationships between tumour sectors (T) and the adjacent non-tumour liver tissues (N), based upon similarities and differences in their genetic characteristics[22]. In general, we observed concordance between immune-ITH and RNA- or DNA-ITH where tumours with low immune-ITH showed shorter RNA or DNA branch distances between tumour sectors compared to those with high immune-ITH (Fig. 2a). Next, we calculated the tumour RNA-ITH, as 1 minus Spearman's $\rho$ for RNA expression of each gene and DNA-ITH, as ratio of the number of unique DNA mutations

**Fig. 1 Significant degree of intratumoural heterogeneity (ITH) in the immune landscapes of HCC. a** Two to five tumour sectors (T), one adjacent non-tumour sector (N) and one PBMC (P) sample were collected from each of the 28 HCC patients and analysed by CyTOF, RNA sequencing (seq) or whole genome sequencing (WGS) for their genomic, transcriptomic and immunomic profiles. **b** Global tSNE plots showing 30 immune clusters (0–29) from all tumour sectors ($n = 95$), each represented by one colour. **c** Heatmap depiction of 30 immune clusters (rows) with normalized protein markers expression (columns) from all samples. The colour bars on the left correspond to the major immune lineages. **d** Graphs showing proportions of 30 immune clusters in each tumour sector from five representative HCC patients. **e** Bar graphs showing proportions of 15 immune subsets (percentages of each immune subsets of total live CD45+ immune cells) in all 95 tumour sectors from 28 HCC patients (each patient labelled and separated by grey colour zone). Bottom, heatmap showing relative immune-ITH scores of 28 HCC patients with respect to its median value. Each bar represents a single tumour sector. **f** Scatter plot showing correlation between immune-ITH scores calculated from tSNE or manually gated immune clusters. **g** Scatter plot showing correlation between immune-ITH scores by Spearman's correlation and Euclidian distance metrics. **h** Left: representative images from multiplex immunohistochemistry (mIHC) stained for CD8 (green), CD4 (red) and DAPI (blue) on either homogenous (patient A002) or heterogeneous (patient C002) tumours. Scale bar, 50 μm. Right: correlation between immune-ITH calculated by Spearson's correlation from CyTOF data and standard deviation (SD) of CD4+ and CD8+ T-cell density derived from mIHC data. **f–h** Spearman's correlation coefficient, $\rho$- and $p$-value were indicated.

to the total number of DNA mutations, with references to previously described methods[9,23] (Supplementary Fig. 4b). Comparing these ITHs, we found a strong correlation between immune-ITH and RNA-ITH and a positive trend with DNA-ITH (Supplementary Fig. 6a), indicating a closer relationship between immune and tumour transcriptome landscapes.

To further illustrate their relationship with tumour mutational landscape, we examined genome-wide alteration fractions or copy number variations (CNVs) in tumours with high vs. low immune-ITH. We observed higher events of CNVs, indicating higher genomic instability in tumours with high immune-ITH (Fig. 2b, c). In particular, we identified 214 deleted and 114 amplified cytobands associated with high immune-ITH (Supplementary Table 3). Several well-known tumour suppressor genes, such as *PTEN*, *NOTCH1*, *APC*, *CDKN2A* and *FAT1*, were among the deleted cytoband, whereas known oncogenes, such as *VEGFA*, *MUC1*, *NTRK1*, *SHC1* and *JTB*, were among the amplified regions (Fig. 2b, d). These mutations are known to be associated with tumour aggressiveness and progression, placing immune-ITH along tumour progressive mutational trajectory. Interestingly, the amplification of VEGF in tumours with high immune-ITH indicated a link to angiogenesis[24] and anti-VEGFA combined with anti-PD-L1 ICB has been a successful phase III trial in advanced HCC (IMbrave150)[5]. One of the deleted genes, *APC*, is a gene from Wnt signalling pathway, which has been shown to be associated to immune exclusion[25]. However, CNV of *CTNNB1*, another gene known to be linked to immune exclusion[25,26], was not significantly related to immune-ITH (Fig. 2d). To show the relationship between immune-ITH and immune exclusion, we compared immune-ITH and tumour tissue T-cell density in our current cohort, and found no significant association between them (Supplementary Fig. 7a, b). We hence propose that immune-ITH is an immune evolutionary event independent of immune exclusion.

Taken together, immune-ITH is linked to multiple tumour evolutionary events, such as enhanced tumour transcriptomic-ITH and increased CNV, particularly in genes related to tumour aggressiveness and progression.

**Immune exhaustive and suppressive TME in tumours with high immune-ITH**. To further underscore the impact of immune-ITH on the overall immune activation status in tumours, we next compared the proportion of key immune subsets in low vs. high immune-ITH tumours according to their median immune-ITH scores. We found that tumours with high immune-ITH were significantly enriched with immunosuppressive/exhausted GB-inactive memory CD4+ T cells, regulatory T cell (Treg), as well as Tim-3+ and PD-1+GB− exhausted CD8+ T cells, conversely, tumours with low immune-ITH were enriched with activated/cytotoxic immune subsets, such as GB+CD45RO+

activated memory CD4+ T cells, CD69+/− natural killer (NK) cells and Tim-3− or PD-1−GB+ activated CD8+ T cells (Fig. 3a). Eight other immune subsets, including naive CD4+ T cells, Lag-3+/−CD8+ T cells and CD14+macrophages, showed no significant enrichment in neither tumour groups (Supplementary Fig. 8a), although CD27− B cells with unknown functions were also significantly enriched in tumours with low immune-ITH (Fig. 3a). This data demonstrated that immune-ITH is linked to distinct immune subsets distribution and hence the overall immune status of TME.

To further validate the immunosuppressive status of tumours with high immune-ITH, we showed that the intratumoural tissue density of Treg was indeed enriched in the tumours with high immune-ITH (Fig. 3b). Next, we also tested the functionality of CD3+ T cells for cytokines production upon Phorbol 12-myristate 13-acetate (PMA)/Ionomycin stimulation and observed varying percentages of cytokine-expressing T cells across different tumour sectors, demonstrating a marked degree of ITH in T-cell functionality (Fig. 3c). Consistent with the findings above, the levels of the pro-inflammatory cytokines tumour necrosis factor-α and interferon-γ in stimulated CD3+ T cells were lower in tumours with high vs. low immune-ITH (Fig. 3d).

Overall, tumours with low immune-ITH experienced stronger immune pressure; while tumours with higher immune-ITH harboured a more immunosuppressive and exhaustive TME. Such gradient towards immune inactivation with increased immune-ITH indicates TME remodelling and immune evolution, which could have important implications in tumour progression.

**Immunoediting events in tumours with different immune-ITH**. Given the link between immune-ITH and intratumoural immune status, we next examined non-silent mutations, loss of heterozygosity in human leukocyte antigens (HLA-LOH) and immunoediting[16], events known to be driven by immunoselective pressure. Comparing low vs. high immune-ITH tumours according to median immune-ITH scores, we observed higher total non-silent mutations in high immune-ITH tumours (Fig. 4a), suggesting a tumour evolutional trajectory with the accumulation of more mutational burden. Next, we examined specific genomic mutations, the neoantigen (8-mer to 11-mer epitopes with <500 nM predicted binding affinity to major histocompatibility complex (MHC) class 1), which could be presented to and recognized by the immune system. Interestingly, tumours with high immune-ITH harboured higher total and subclonal (occurred in at least one but not all sectors), but not clonal (occurred in all sectors) neoantigens (Fig. 4b). This data suggests that the heterogeneity of neoantigens across tumour sectors represented by higher subclonal neoantigens mirrored that of the increased immune-ITH. Also as genomically heterogenous tumours with higher subclonal neoantigens was shown to

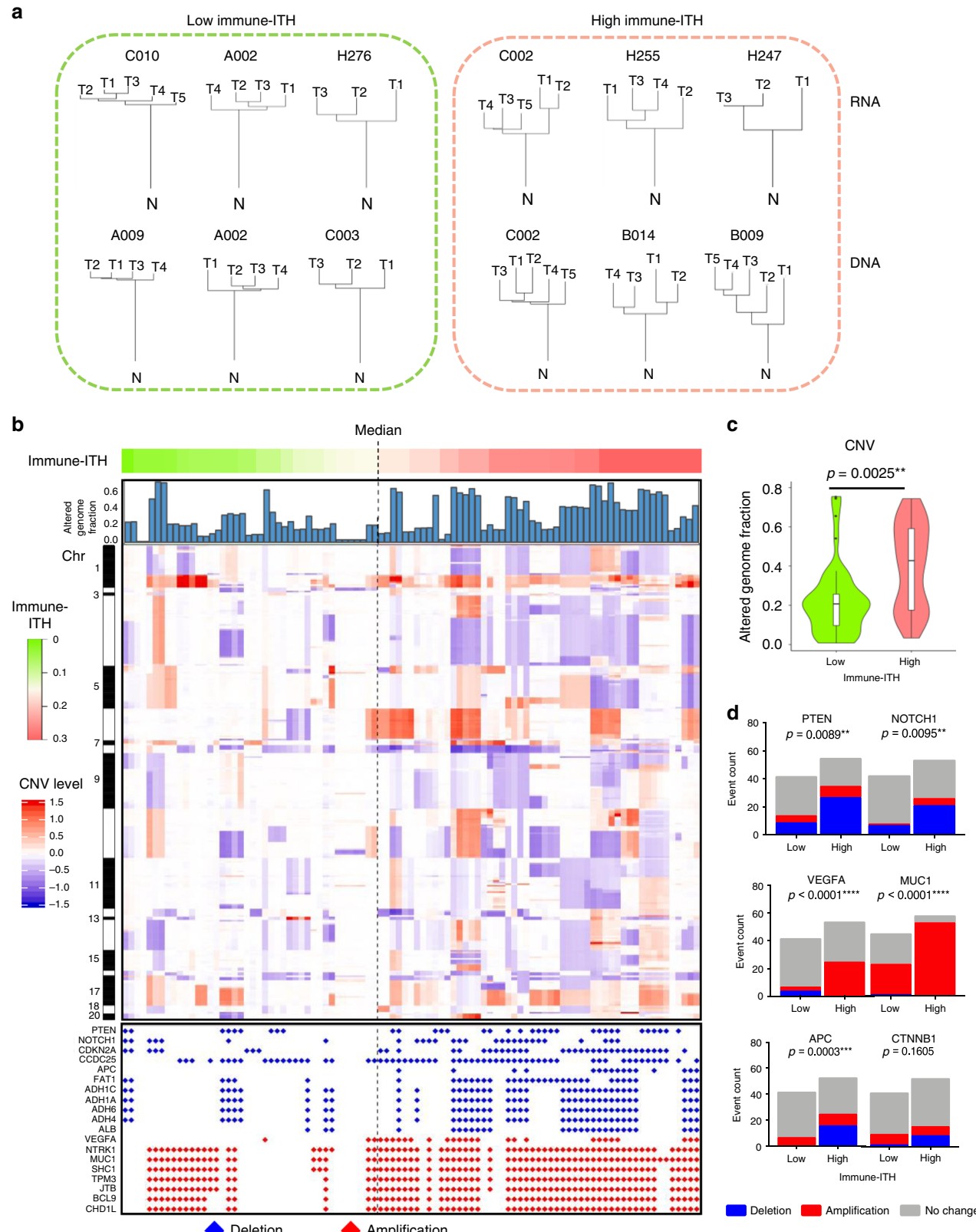

be an indication of tumour evolution[27], this again provided evidence of potential co-evolution between the tumour and immune landscapes, where both demonstrated enhanced heterogeneity.

Given that the stronger immunoselective pressure experienced by tumours with lower immune-ITH could serve as a driving force for tumour escape mechanisms such as HLA-LOH and immunoediting[16], we next mapped the HLA-LOH and immunoediting (represented by the ratio of neoantigen/non-silent mutation) against immune-ITH from each tumour sector (Fig. 4c). Interestingly, we observed lower immune-ITH in tumours with HLA-LOH (Fig. 4d), indicating a defective antigen-presentation machinery (HLA-LOH) as a tumour escape

**Fig. 2 Tumour evolution trajectory along immune-intratumoural heterogeneity (ITH). a** Representative RNA and DNA phylogenetic trees from tumours with low vs. high immune-ITH. The genetic distances between each tumour sector (T), as well as between tumours and adjacent non-tumour tissue (N), were represented by the length of the tree branches drawn to scale. **b** Comparing low vs. high immune-ITH across 95 tumour sectors (top), bar chart showing the altered genome fraction (second top), heatmap showing differential copy number variations (CNV) level (middle) and copy number status of the selected genes showing significant deletion or amplification (bottom). Source data are provided as a Source Data file. **c** CNV by altered genome fractions in low vs. high immune-ITH tumours. **P < 0.01 by two-sided Wilcoxon's rank-sum test. **d** Selected genes showing significant deletion, amplification or no change respectively, between tumours with low vs. high immune-ITH. Two-tailed P-values by Fisher's exact test, **P < 0.01, ***P < 0.001, ****P < 0.0001.

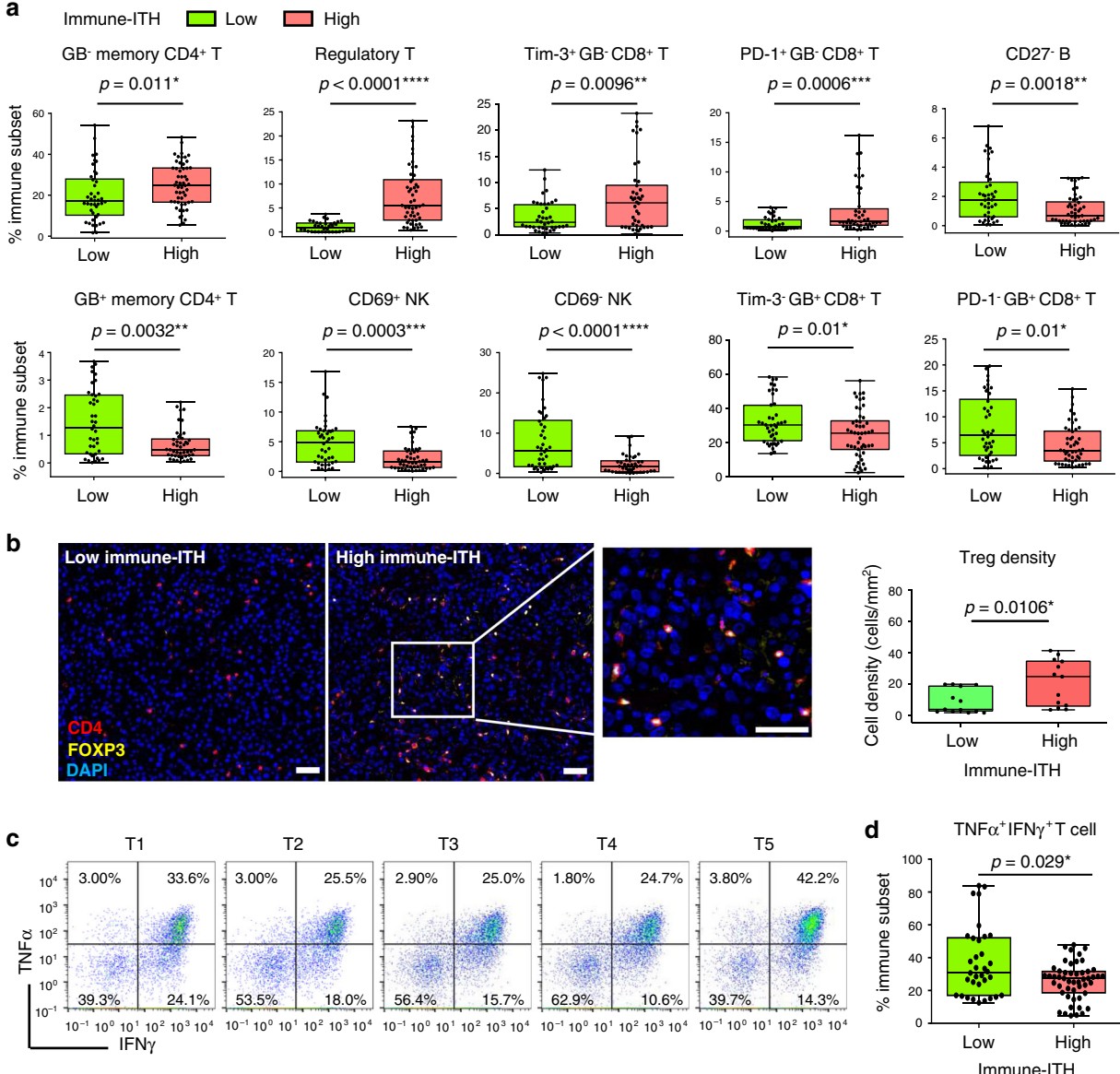

**Fig. 3 Differential immunoselective pressure in tumours with low vs. high immune-intratumoural heterogeneity (ITH). a** The immune subsets that were significantly enriched in tumours with low vs. high immune-ITH (n = 95 total tumour sectors). **b** Left: representative multiplex immunohistochemistry (mIHC) images stained for CD4 (red), Foxp3 (yellow) and DAPI (blue) on tumour tissues with low (patient B016) or high (patient H319) immune-ITH. Scale bar, 50 μm. Right, Treg cell density (count/mm²) in tumours with low vs. high immune-ITH (n = 26 tissues). **c** Dot plots showing percentages of intracellular pro-inflammatory cytokines IFNγ and TNFα in CD3⁺ T cells from T1 to T5 tumour sectors of representative Patient H319. **d** Percentage of TNFα and IFNγ-expressing CD3⁺ T cells in tumours with low and high immune-ITH (n = 95 total tumour sectors). Cells were stimulated with PMA/ Ionomycin for 5 h. **a, b, d** Data were shown by box plot. The whiskers represent minimum and maximum values, the band inside the box is the median and box edges show the first and third quartiles. *P < 0.05, **P < 0.01, ***P < 0.001, ****P < 0.0001 by two-sided Mann–Whitney U-test.

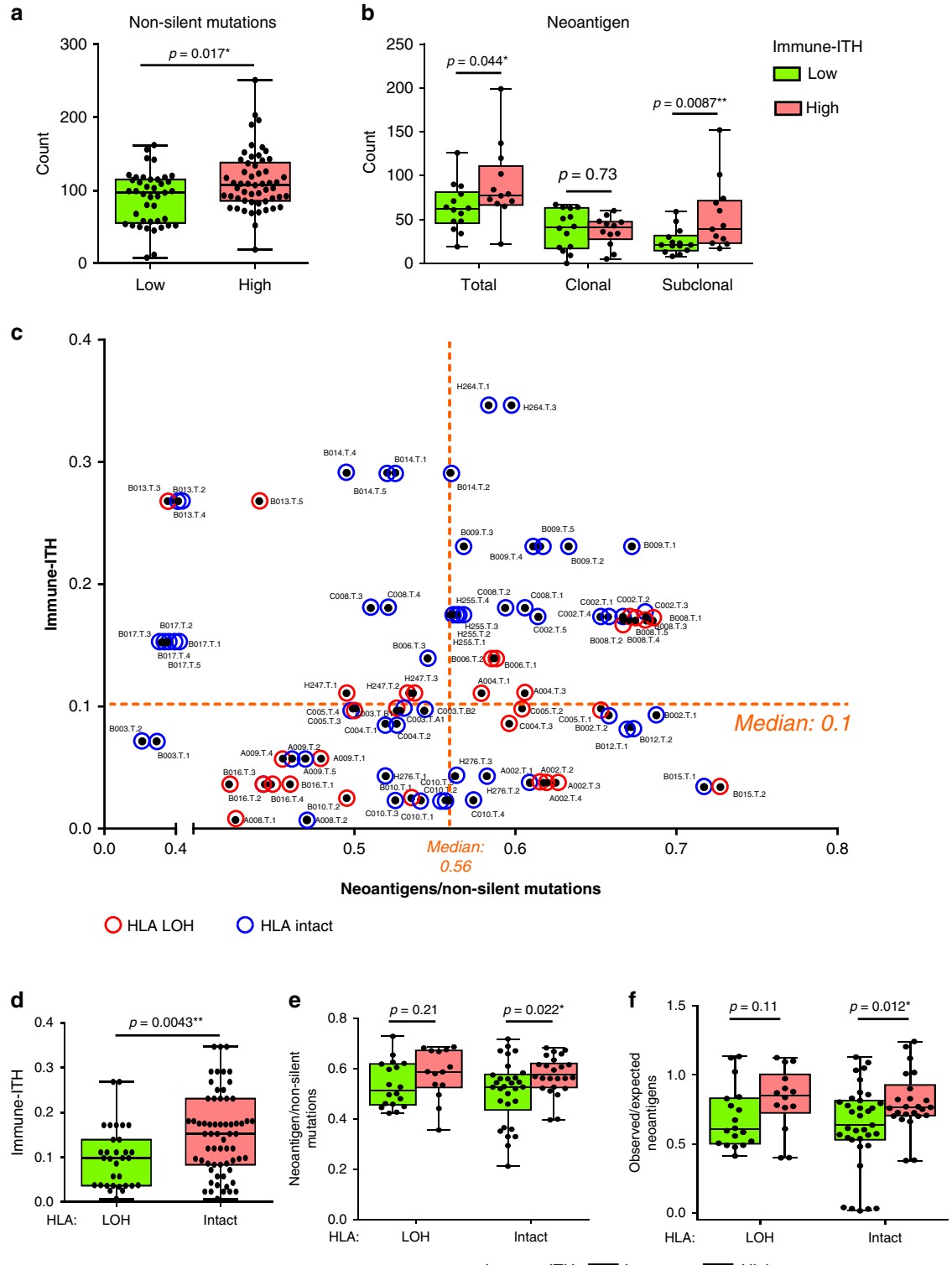

**Fig. 4 Significant HLA-LOH and immunoediting in tumours with low immune-ITH. a** Total non-silent mutational load in high vs. low immune-ITH tumours (*n* = 95 total tumour sectors). **b** Total, clonal and subclonal neoantigen loads in low vs. high immune-ITH tumours (calculated based on 95 tumour sectors from 28 patients). **c** Immune-ITH scores of each HCC tumour sector were plotted against its neoantigen/non-silent mutations ratio, which is indicative of immunoediting. Orange dash line denotes median level of immune-ITH or median of neoantigen/non-silent mutations, as indicated respectively. Red circles show tumours with HLA-LOH event and blue circles show intact HLA. **d** Immune-ITH scores in tumours with or without HLA-LOH. **e** Ratio of neoantigen over non-silent mutation and **f** ratio of observed : expected neoantigen in high vs. low immune-ITH tumours with or without HLA-LOH. **d**-**f** *n* = 95 total tumour sectors. **a**, **b**, **d**-**f** Data were shown by box plot. The whiskers represent minimum and maximum values, the band inside the box is the median and box edges show the first and third quartiles. *P < 0.05 or **P < 0.01 by two-sided Mann–Whitney *U*-test.

mechanism in tumours with low immune-ITH. In addition, we observed significant immunoediting (indicated by lower neoantigen/non-silent mutation ratio) only in tumours with low immune-ITH and intact HLA (Fig. 4e), indicating that only tumours with stronger immune pressure (low immune-ITH) and antigen-presentation capability (intact HLA) underwent immunoediting as another escape mechanism. We further confirmed this by calculating the immunoediting score as the ratio of observed : expected neoantigens per non-silent mutation according to previously published methods[16] and showed significant immunoediting event only in low immune-ITH tumours with intact HLA (Fig. 4f). This data strengthens the link between immune-ITH, immunoselective pressure and tumour escape mechanisms as a series of tumour evolutionary events.

Taken together, tumours with low immune-ITH experienced higher immunoselective pressure and underwent HLA-LOH and immunoediting as the intrinsic tumour escape mechanisms. Whereas tumours with high immune-ITH escaped with extrinsic mechanisms by remodelling towards a more immunoexhaustive and suppressive TME as well as accumulated more mutations particularly subclonal neoantigens. These distinct evolutionary mechanisms provided evidence for tumour-immune co-evolution along the gradient of immune-ITH in HCC tumours.

**Transcriptomic signature and networks of immune-ITH.** To provide mechanistic insights to immune-ITH, we next examined the transcriptomic or molecular signatures associated with immune-ITH. We first identified a total of 1709 differentially expressed genes (DEGs) as the 'immune-ITH transcriptomic signature' when comparing tumours with low vs. high immune-ITH (Supplementary Fig. 9a). To further explore the interactions between the tumour-specific transcriptome with its immune landscapes, we filtered off from these DEGs the genes contributed by immune subsets according to CIBERSORT[28] to obtain the tumour-specific transcriptomes. We then correlated these genes with the proportion of key immune subsets in TME (from CyTOF) and found very distinct transcriptome-immune networks between tumours with low vs. high immune-ITH (Fig. 5a, b). Tumour transcriptome from low immune-ITH showed positive association with cytotoxic NK cells and activated GB$^+$memory CD4$^+$ and PD-1$^-$ GB$^+$ CD8$^+$ T cells, or negative association with immunosuppressive Treg, exhausted PD-1$^+$GB$^-$CD8$^+$ T cells and inactive GB$^-$ memory CD4$^+$ T cells (Fig. 5a). The opposite transcriptome-immune interactions were observed in high immune-ITH, whereby positive correlations with immunosuppressive and exhaustive or negative correlations with cytotoxic and activated immune subsets were observed (Fig. 5b). These networks demonstrated that transcriptomic signature of high immune-ITH was closely linked to an immunosuppressive and exhausted TME, consistent with our data above.

Next, we performed pathway enrichment analyses on the DEGs from these low or high immune-ITH networks (Fig. 5c and Supplementary Table 4). We found that metabolism pathways, particularly genes associated with fatty acid metabolism, such as *CBR4*, *CPT2*, *ACAA2*, *ECHDC2* and *ACAA1*, as well as glycoprotein-related genes such as *CLEC1A*, *ADAMTS4*, *COL25A1*, *CLEC3B*, *ADAMTS1* and *CLEC2B* were enriched in tumours with low immune-ITH (Fig. 5c, d), indicating the distinct metabolism pathways were potentially involved in maintaining the immune status of TME in HCC. Indeed, metabolic regulation of immune functions in cancer have been increasingly appreciated in a number of recent studies[29,30].

Conversely, the pathways enriched in trancriptome-immune network from tumours with high immune-ITH included cell cycle, nucleotide-binding, centromere, microtubule and transcription (Fig. 5c), all of which were well known to be associated with tumour cell proliferation and disease progression[31–33]. In particular, cell cycle genes such as *CSNK2A1*, *MCM8*, *CDK19*, *KIFC1* and *MCM7* (Fig. 5d), among which *KIFC1* was previously found to be a factor for poor prognostic and a therapeutic target associated with tumour proliferation in HCC[34,35], even though its immunodulatory function has never been described before. Another group of chaperone genes, such as *CCT6A*, *CCT4*, *TCP1*, *CCT5*, *PTGES3* and *CCT7* (Fig. 5d), were previously implicated in cancer cell proliferation and predicts poor prognosis in HCC[36,37].

The distinct gene-immune networks further strengthened the evidence that transcriptomic signature of immune-ITH was closely linked to the phenotypes of its TME. More importantly, it demonstrated that tumours with high immune-ITH formed a transcriptomic network linked to immune exhaustion or suppression, as well as tumour proliferation and disease progression.

**High immune-ITH predicts worse disease prognosis and survival in HCC patients.** Although genomic- and transcriptomic-ITH has been shown to correlate with poor prognosis in various cancers[38], the clinical relevance of immune-ITH remains unknown. Our data above suggest that tumours with high immune-ITH are more immunosuppressive, harboured more mutations and show aggressive or progressive tumour transcriptomic signature. By examining immune-ITH against multiple clinical parameters, we indeed found that high immune-ITH was associated with larger size of tumours, higher degree of fibrosis, the presence of microvascular invasion (MVI) and advanced tumor node metastasis (TNM) stage of tumour (Fig. 6a, b), all of which indicative of tumour progression and poor disease profiles. Of note, the immune-ITH is not associated with other parameters such as grade, alpha fetoprotein (AFP) level or the number of tumour sectors collected and analysed (Supplementary Fig. 10a). It is also independent of the viral hepatitis status of the patients (Supplementary Fig. 10b).

More importantly, patients with tumours of high immune-ITH had a significantly higher risk of recurrence than those with tumours of low immune-ITH (Fig. 6c). Of note, patients received no treatment prior to the point of recurrence, showing this as a phenomenon following the natural trajectory of tumour evolution and disease progression that was not influenced by any therapeutic intervention. Next, we examined the survival impact of immune-ITH in our current cohort with both univariate and multivariate analyses, taking into consideration of multiple clinical factors including tumour stage, grade, size, MVI, as well as other factors associated with immune-ITH including cytokine-expressing CD3$^+$ T cells (representing TME immunoselective pressure), DNA- and RNA-ITH, and tumour neoantigen burden. Among all the parameters, only immune-ITH as well as stage, tumour size, MVI and race were significantly linked to recurrence-free survival (RFS) in the univariate analysis (Supplementary Table 5). From the multivariate analysis, we found that only immune-ITH remained an independent predictive factor for RFS, together with stage and MVI (Fig. 6d). Hence, to rule out potential confounding effect from both stage and MVI, we tested the impact of immune-ITH only in patients with tumours from early stages (TNM stage I and II) or without MVI and found that immune-ITH remained an independent predictor of RFS (Fig. 6e). Taken together, increased immune-ITH was significantly associated with worse clinical profile and predicts for poor disease outcome in HCC patients.

Lastly, to validate the impact of immune-ITH on larger publicly available HCC dataset, we interrogated the expression

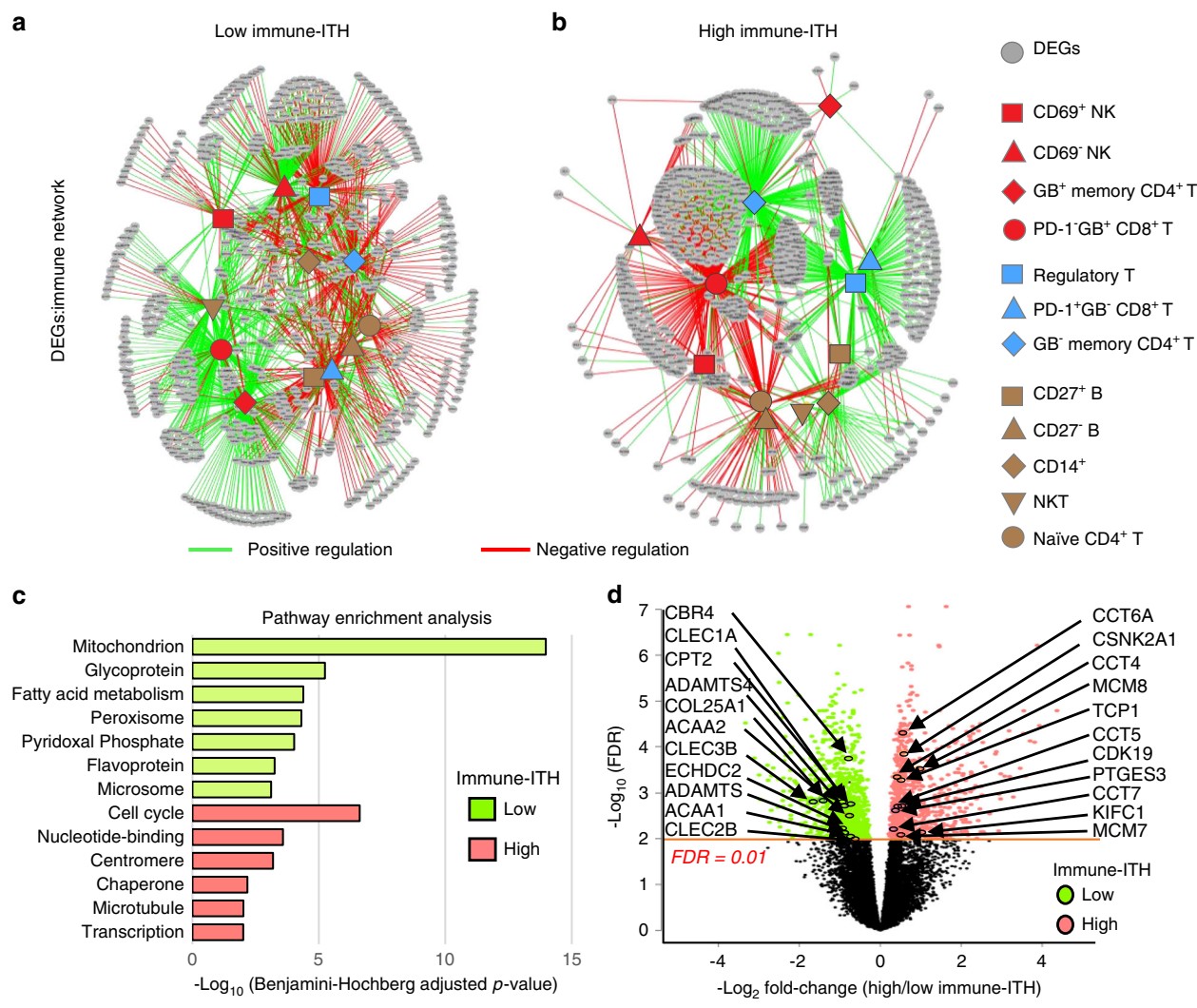

**Fig. 5 Transcriptome-immune networks associated with immune-intratumoural heterogeneity (ITH). a**, **b** Transcriptome-immune network showing correlation between the immune subsets and the tumour transcriptome associated with **a** low or **b** high immune-ITH. Green and red lines denote positive and negative correlation, respectively (Spearman's correlation test, $\rho \geq 0.4$, $P < 0.05$). The immune subsets are denoted as red: cytotoxic/activated; blue: immunosuppressive/exhausted; and brown: controversial roles. **c** DAVID pathway enrichment analysis of the genes enriched in tumours with low or high immune-ITH. **d** Volcano plot showing DEGs with selected genes as highlighted in the low and high immune-ITH tumours. The orange line denotes the false discovery rate (FDR) = 0.01. Source data are provided as a Source Data file.

profile of 1709 immune-ITH gene signature in two large public HCC datasets: the Japanese Liver Cancer from the International Cancer Genome Consortium[39] ('Japanese,' $n = 203$) and the Liver HCC from TCGA[40] ('TCGA,' $n = 315$). For each dataset, we clustered the patients into low or high immune-ITH associated groups based on significant differential expression of immune-ITH related genes (Supplementary Fig. 11a). Indeed, we confirmed that patients with gene expression profiles resembling high immune-ITH had poorer overall survival (OS) than those with gene expression profiles resembling low immune-ITH, in both the Japanese and TCGA cohorts (1000-time bootstrap false discovery rate (FDR) < 0.01, Fig. 6f). Therefore, despite the fact that these publicly available transcriptomic data were obtained from single tumour biopsy, the molecular features underlying immune-ITH were consistently and closely linked to advanced clinical trajectory. We further confirmed the robustness of our observations by shifting one patient between the immune-ITH groups and using leave-one (patient) out method to show that >95% of target genes remained consistent and capable of segregating patients' OS (Supplementary Fig. 11b, c). Hence, we

have identified and validated a robust immune-ITH signature capable of predicting disease prognosis in HCC patients.

In conclusion, we proposed a tumour-immune co-evolution model (Fig. 6g), where the immune landscapes evolved with increased immune-ITH and immunosuppressive/exhaustive TME; concurrently, the tumours evolved with accumulation of more mutations and escaped using HLA-LOH or immunoediting. Collectively, these events lead to tumour progression and early recurrence, making immune-ITH a hallmark of tumour evolution and progression.

## Discussion

The clinical relevance of immune-ITH and its relationship with tumour evolution were not explored previously. Our current study uncovered significant degree of immune-ITH from multiple HCC tumour regions and their inter-relationships with tumour evolution and impact on clinical outcome. We observed a significant degree of immune-ITH, which is linked to transcriptomic-ITH. Importantly, a gradient of immunoselective

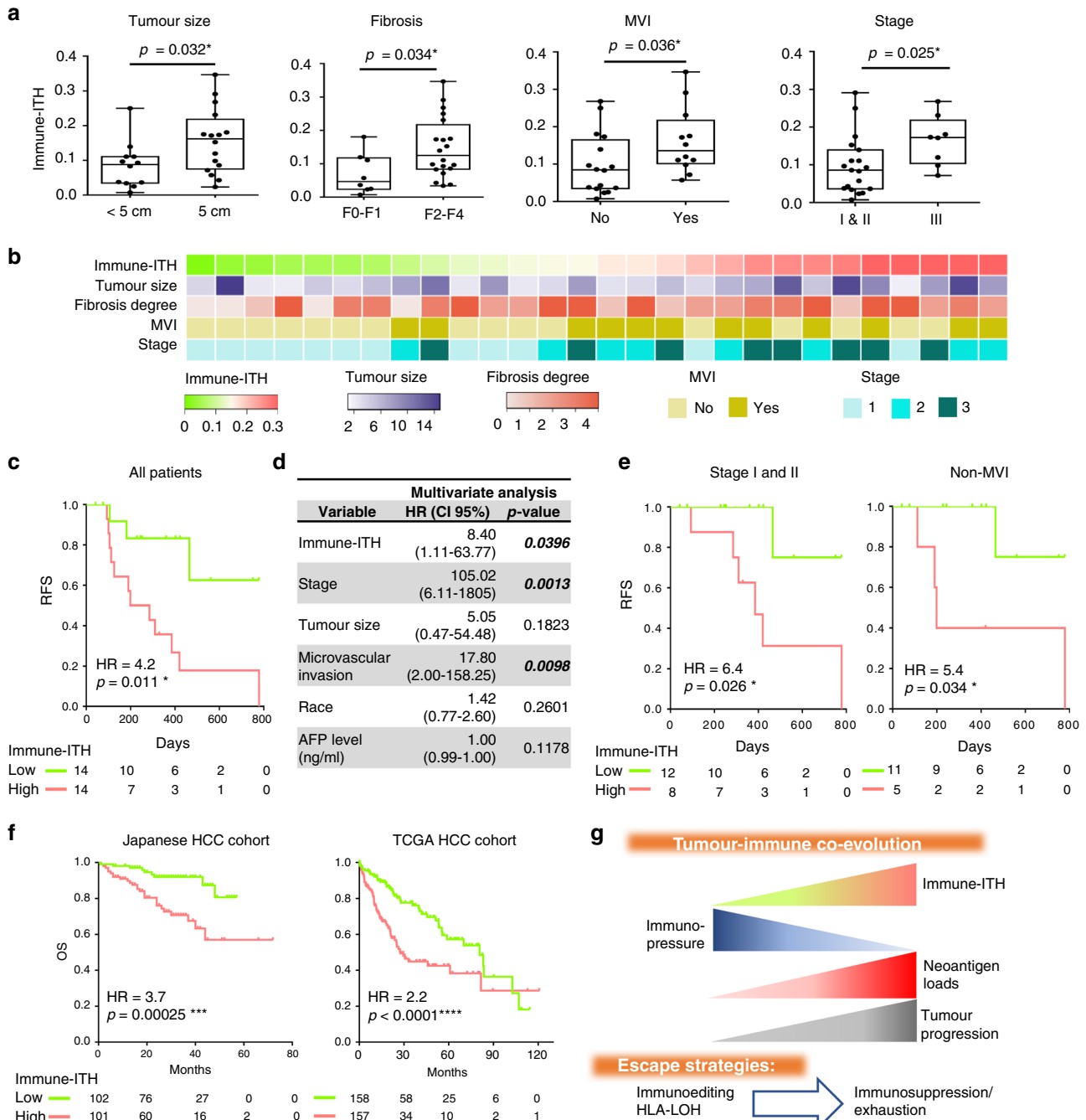

**Fig. 6 High immune-intratumoural heterogeneity (ITH) is associated with poor patient clinical outcome. a** Immune-ITH in tumours: <5 cm vs. ≥5 cm (size based on Milan criteria); low (F0-F1) vs. high (F2-F4) fibrosis stage (METAVIR scoring system), without or with presence of microvascular invasion (MVI) and early (stage I and II) vs. late (Stage III) stage (TNM version 8). Data were shown by box plots from 28 patients. The whiskers represent minimum and maximum values, the band inside the box is the median and box edges show the first and third quartiles. *P < 0.05 by two-sided Mann–Whitney *U*-test. **b** Heatmaps showing immune-ITH, tumour size (cm), microvascular invasion (yes as present; no as absent) and Stage (TNM version 8) from 28 patients. **c** Kaplan–Meier curves for recurrence-free survival (RFS) profiles of 28 HCC patients with low or high immune-ITH tumours. **d** Multivariate analysis of clinical and biological variables using Cox proportional hazards regression models (*n* = 28). **e** Kaplan–Meier curves for RFS profiles of patients with tumours from early stages (*n* = 20) or without MVI (*n* = 16) segregated by low or high immune-ITH tumours. **f** Kaplan–Meier curves for overall survival (OS) profiles of Japanese (*n* = 203) and TCGA cohorts (*n* = 315) segregated DEGs associated with immune-ITH. Kaplan–Meier graphs showing immune-ITH low (green) or high (red). **c, e, f** Kaplan–Meier graphs showing immune-ITH low (green) or high (red); Hazard ratio (HR) and log-rank test two-tailed *P*-values as indicated. **g** A graphical summary of tumour-immune co-evolution model with distinct tumour escape strategies.

pressure was uncovered along with immune-ITH, under which the tumours employed distinct mechanisms to escape accordingly. The immune-ITH transcriptomic signature provided insights in pathways associated with tumour progression and dampening of immune response in TME. Lastly, our data demonstrated that immune-ITH correlates to worse clinical profile and could predict for poorer prognosis in HCC patients, emphasizing immune-ITH as a hallmark of tumour evolution and critical indicator of disease progression. Taken together, our current findings show extensive cross-talk between tumour and

immune microenvironments, which co-evolve along tumour progression with increased immune-ITH, enhanced TME exhaustion and tumour mutations, supporting a tumour-immune parallel evolution model (Fig. 6g).

Despite previous work that described the heterogenous landscapes of HCC[6,11,14], the impact of immune-ITH is largely unknown. Ma et al.[6] employed single cell RNA sequencing to analyse the multiple tumour sectors before ICB and concluded that tumour cell transcriptomic diversity is linked to clinical response to ICB. However, the biological and clinical relevance of immune-ITH remain unexplored. Losic et al on the other hand focused mainly on how T cell receptor (TCR) clonal heterogeneity is correlated to tumour genomic ITH in liver cancer[14], while our study focused on the heterogenous composition and spatial locality of TILs as an evolutionary event that is relevant to clinical outcome. Lastly, despite the fact that the immune landscapes from multiple regions of HCC were previously analysed, the tumours were segregated into high or low immune infiltration for correlation with clinical outcome[11], which again did not demonstrate the relevance of immune-ITH as an an evolutionary event of tumour progression as we have shown herein.

It is important to appreciate that tumour-immune interaction is heterogeneous, dynamic and also bi-directional. For instance, immune pressure could potentially drive tumour genomic evolution[16,41]; in return, immune landscapes are also constantly being shaped by the tumour transcriptomic landscapes[42,43]. From our current data, the HCC TME shifted from homogenously 'good' to heterogenously 'bad' albeit exhausted and suppressive TME, forming a gradient of decreasing immunoselective pressure, an indication of immune landscape remodelling and evolution. The tumours, on the other hand, accumulated more mutations especially subclonal neoantigens, showing its parallel evolution trajectory. Hence, as concluded in the current study, the tumour-immune dynamic is changing constantly to adapt to one another, strengthening the concept of tumour-immune co-evolution championed by several previous studies[44]. Our findings also highlight that even within a single tumour, each tumour sector harbours its own unique mutation and microenvironment. This poses a notable challenge to conventional clinical decision-making, which is based on sampling of a single tumour biopsy. Despite a recent report that claimed the reliability of single-sample in HCC from multi-region sampling and analysis using mainly IHC[45], our data proposed herein the existence of significant immune heterogeneity with clinical relevance with in-depth immuno-phenotyping.

As both tumour mutational burden and TME have important implications in the response to immunotherapy[46,47], higher total mutational burden, neoantigens loads as well as higher frequency of exhausted PD-1+CD8+ T cells (prime target for anti-PD-1 ICB) in HCC tumours with high immune-ITH could potentially show better respond to immunotherapy. However, it must also be taken into consideration that these tumours are also infiltrated with more Treg and harboured higher subclonal neoantigen levels (consistent with a more heterogenous tumour), which may dampen the response and leading to resistance to immunotherapy[48,49]. Moreover, the recent study from liver cancer patients treated with ICB demonstrated that low tumour transcriptomic diversity and higher cytolytic activity of CD8+ T cells, consistent with our tumours with low immune-ITH, predicts clinical response to immunotherapy[6]. On the other hand, our current data show VEGFA amplification in the tumours with high immune-ITH, indicating the potential therapeutic benefit of combining anti-angiogenesis agent and checkpoint blockade in advanced HCC as demonstrated in recent successful phase III trial (IMbrave150) using atezolizumab (anti-PD-L1) and bevacizumab (anti-VEGFA)[5]. Therefore, it will be very important and interesting to study how this immune-ITH could affect response to immunotherapy with a deeper understanding of this intratumoural immune–host dynamics, which we believe would be helpful to stratify HCC patients for precision immunotherapy.

In conclusion, our study deciphers the complexity of intratumoural immune–host interaction and provides evidence showing immune-ITH as a hallmark of tumour-immune co-evolution along HCC progression.

## Methods

**Patients.** Ninety-five tumour sectors from two to five regions per tumour (T), 28 matched adjacent non-tumour liver tissues (N) and pre-surgical blood were obtained fresh from 28 HCC patients underwent surgical resection at Singapore General Hospital, National Cancer Centre Singapore and National University Hospital (Supplementary Table 1). This is part of the on-going Prospective Cohort Study on the Clinical Trajectory of Resected Hepatocellular Carcinoma (PLANET) (NCT03267641), an observational cohort study following standard of care- liver cancer resection and routine follow-up with primary outcome measured as time to recurrence. The study was approved by the Central Institution Review Board (CIRB) of SingHealth, of which all National Cancer Center Singapore, Singapore General Hospital and National University Hospital were constituent members (CIRB Ref: 2016/2626 and 2018/2112). Each patient gave informed written consent. Patients received no pre- or post-surgical treatment until recurrence, consistent with current standard of care. This allows us to study the natural progression of disease without the influence from treatment. Patients were monitored prospectively with regular imaging and other clinical investigations. Strict protocol of multi-sector tumour sampling was followed (Supplementary Fig. 1a, b) and each sector was divided for CyTOF, WGS and RNA sequencing (Fig. 1a). TILs and non-tumour tissue-infiltrating leukocytes (NILs) were isolated by enzymatic digestion and peripheral blood mononuclear cells (PBMCs) by Ficoll-Paque layering[17].

**Cytometry by time-of-flight.** TILs, NILs and PBMCs were either unstimulated or stimulated with PMA and ionomycin (Sigma). Cells were processed and stained with 38 antibodies (Supplementary Table 2) purchased preconjugated or conjugated in-house according to the manufacturer's instructions (Fluidigm) before analysis on a HeliosTM mass cytometer (Fluidigm). The generated files were analysed by FlowJo (v.10.2; FlowJo): live single cells (cisplatin-negative and DNA-intercalator-positive) were debarcoded to each sample file based on their unique CD45 barcodes. Each file was then down-sampled to 10,000 cells and further analysed using our in-house The Extended Polydimensional Immunome Characterization (EPIC) analysis pipeline, which contains the browser-based R Shiny app 'SciAtlasMiner'[19] for data visualization. Clustering was performed using the Phenograph (v.1.5.2)[18] algorithm that automatically determines the number of clusters. Dimension reduction was carried out using fast interpolation based t-distributed neighbour embedding (fi-tSNE, v.1.0.1). Manual gating of 15 immune subsets was performed with FlowJo (v.10.2).

**RNA sequencing.** Total RNAs were isolated from tissues using Picopure RNA-Isolation kit (Arcuturus, Ambion) and cDNAs constructed using SMART-Seq®v4 UltraTM Low Input RNA Kit (Clontech, USA). Illumina-indexed libraries were generated using Nextera-XT DNA-Library Prep Kit (Illumina, USA). RNA sequencing was performed on HiSeq High output platform at the Genome Institute of Singapore (GIS).

The raw reads were aligned via STAR v2.5.2a[50] to the Human Reference Genome hg19 and the expected gene-level counts were calculated using RSEM v1.3.0[51]. Only protein-coding genes with >1 count/million reads in ≥5% of the samples were retained. Data were normalized using DEseq2 v1.22.2 and DEGs analysis was performed using the R package v3.4.4 with Limma v3.38.3[52] at FDR < 0.01. Pathway enrichment analyses were performed using DAVID v6.8. For RNA-immune network and correlation analyses, genes from immune subsets were filtered out according to the gene list provided by CIBERSORT[28] and correlation plots were generated using ggplot2 v3.1.1 for selected DEGs with $p < 0.05$ and $\rho \geq 0.4$.

**Whole genome sequencing.** DNA was extracted from tissues using Qiagen All-Prep kit, DNA fragments were end-repaired, ligated with sequencing adapters, amplified, and sequenced by Illumina sequencing platform at GIS. Raw reads were mapped to the Human Reference Genome hg19 using Burrows–Wheeler Aligner v0.7.12. Duplicated reads were removed, base-quality recalibration and realignment were performed using Genome Analysis Tool Kit v3.1. Somatic variants were called by comparing tumour vs. non-tumour samples using Mutect v1.1.7[53].

**ITHs quantification and immunohistochemistry.** The proportions of immune clusters from tSNE analysis or manual gating (Supplementary Fig. 2a) in each tumour sector were calculated. The ITH scores were calculated for patients with ≥2 tumour sectors using pairwise comparison of all sectors for: (i) DNA, as ratio of the number of unique DNA mutations to the total number of DNA mutations; (ii) RNA, as 1 minus the Spearman correlation coefficient's $\rho$ of RNA expression of each gene; and (iii) immune, as 1 minus the Spearman's $\rho$ of proportions of the

immune subsets. The median values were taken as the patient-level ITH scores. Phylogenetic trees were constructed based on the distance matrix defined between the samples using the neighbour-joining algorithm[54]. For DNA, we used the hamming distance between mutational profiles of the samples. For RNA, we used 1 − Spearman's correlation as the distance measurements between samples.

Multiplex IHC (mIHC) on representative formalin-fixed paraffin-embedded tissues ($n = 26$), was performed with anti-human CD4 (Abcam, clone EPR6855, 1 : 200), CD8 (DAKO, clone C8/144B, 1 : 200) and Foxp3 (Abcam, clone 236 A/E7, 1 : 100) antibodies using Opal™ 7-Color IHC Kit (Perkin Elmer) according to the manufacturer's instructions. The density of CD4+ and CD8+ was quantified as number of cells/mm² from $10 \times 3mm^2$ representative fields. We then calculated the SD of cell density across tumour regions, which reflects the heterogeneity or similarity, for each tumour. The degree of fibrosis were also scored according to Metavir scoring system[55] using standard haematoxylin and eosin staining on the adjacent non-tumour liver tissue sections.

**CNV analysis and non-silent mutations**. Somatic CNVs were called and segmented with Sequenza v2.1.2. Gene-level copy numbers were obtained with GISTIC(v2.0) using segmented copy numbers[56]. Altered genome fraction was calculated by considering the segments with integer copy number greater or less than the median copy number. The fraction was calculated as total length of aberrant segments/total length of all segments. To compare the altered genome fractions in two immune-ITH groups, a two-sided Wilcoxon's rank-sum test was used. The CNV frequency differences were identified by comparing the cytoband level copy numbers between two immune-ITH groups using Fisher's exact test with adjusted Benjamini–Hochberg p-values.

Non-silent or nonsynonymous mutations were computed from mutations resulting in both missense (mutations in a single nucleotide that result in alternation in amino acid encoded) and nonsense (mutations in the DNA sequence resulting in a stop codon) mutations[53].

**HLA-LOH analysis, neoantigen prediction and immunoediting scoring**

*MHC class I*. HLA-A, -B, -C genes were determined by Polysolver v1.0[57], purity and ploidy values by Sequenza v2.1.2[58] and HLA copy number calling by LOHHLA[59]. Minor allele copy number < 0.5 was considered as HLA-LOH. Neoantigen prediction was performed using personalized Variant Antigens by Cancer Sequencing (pVacSeq v4.0.10)[60] and variants calling information were obtained from MuTect v.1.1.7[53]. The variant calls were annotated using VEP v86: 8- to 11-mer epitopes with <500 nM binding affinity. Total, clonal (expressed by all tumour sectors) and subclonal (expressed by at least one but not all sectors) neoantigens were computed. The immunoediting scores were computed as the ratio of observed neoantigen/expected mutations as previously described[16]. Briefly, the expected number of non-silent mutations and neo-peptides were calculated based on mutational spectra estimated empirically[16] and compared to the observed number of non-silent mutations and neoantigens.

**Survival analysis**. Kaplan–Meier analysis of RFS was performed with the log-rank (MantelᴇCox) test (GraphPad Prism v7). Univariate and multivariate analyses were performed using Cox proportional hazards model.

Two public HCC datasets were analysed to assess the survival impact of the immune-ITH (using $n = 1,709$ DEGs) and FDR was estimated by 1000-time bootstrap. Specifically, we segregated the patients based on the differential expression level of 1709 DEGs associated with immune-ITH from our current cohort, i.e., the patients showing high expression of low immune-ITH-associated genes was grouped as low immune-ITH group and those with high expression of high immune-ITH associated genes was grouped as high immune-ITH group (Supplementary Fig. 8a). We then examined the survival profile between these two groups of patients by Kaplan–Meier. The raw counts for the Japanese Liver Cancer from the International Cancer Genome Consortium (Liver Cancer-RIKEN, Japan; Project Code LIRI-JP) and the TCGA dataset (Liver HCC, The Cancer Genome Atlas) were downloaded from the International Cancer Genome Consortium Data Portal[39] and FireBrowse[40], respectively. Only protein-coding genes with fragments per kilobase of exon model per million reads mapped >1 (Japanese dataset) or raw counts > 1 (TCGA dataset) in ≥5 samples were retained, and data were normalized using DEseq2.

**Reporting summary**. Further information on research design is available in the Nature Research Reporting Summary linked to this article.

## Data availability

Sequence data used in the study has been deposited at the European Genome–phenome Archive (EGA), which is hosted by The European Bioinformatics Institute (EBI) under the accession code: EGAS00001003814. The remaining data are available within the Article, Supplementary Information and Source data provided with this paper.

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

## Acknowledgements

We thank all members of TII, the clinical research coordinators from NCCS, SGH, NUHS for their contributions and Insight Editing London for language editing of this manuscript. This work was supported by the National Medical Research Council (NMRC), Singapore (ref numbers: NMRC/TCR/015-NCC/2016, NMRC/CIRG/1460/2016, NMRC/CSA-SI/0013/2017, NMRC/CSA-SI/0018/2017, NMRC/OFLCG/003/2018, NMRC/STaR/020/2013, NMRC/CG/M003/2017, LCG17MAY004 and NMRC/OFIRG/0064/2017) and National Research Foundation, Singapore (ref. number: NRF-NRFF2015-04). We thank all the patients who participated in The PLANET study—a prospective cohort study on the impact of intra-tumoral genomic heterogeneity on the clinical trajectory of resected HCC (https://clinicaltrials.gov/ct2/show/NCT03267641).

## Author contributions

P.N. obtained and analysed data, and prepared the paper. S.M. obtained and analysed the genomic/transcriptomic data, and prepared the paper. C.P., N.K. and H.L. analysed the genomic/transcriptomic data. C.J.L. and J.Q.L. processed the samples and performed experiments. M.W. and L.L. provided analytical and technical help for CyTOF. W.L.T. provided genomic/transcriptomic analysis support. T.K.H.L., W.K.W., T.L., W.Q.L. and P.Y.H. prepared and provided tissue samples, and discussed the data. C.Y.C., S.Y.L., P.C. C., H.C.T., S.I., A.K., Y.Y. D., A.C. and B.K.P.G. recruited patients, provided samples and discussed the data. F.G. provided analysis support for immunomics data. S.A. designed the CyTOF pipeline and discussed the data. P.K.H.C. initiated the project, provided patient samples and discussed the data. W.Z. designed and led the genomics analysis and prepared the paper. V.C. designed and led the study, performed the immunomics analysis and prepared the paper.

## Competing interests

The authors declare no competing interests.
