## [Peer Review File · Nature Communications]

Reviewers' comments:

Reviewer #1 (Remarks to the Author): with expertise in immune landscape and HCC

In this paper, the author analyzed multi-omics data consisting of WGS, bulk RNA seq and, CyTOF from multi-region samples within the same HCC tumor from a 28 HCC patient cohort to reveal the relationship between intratumoural immune heterogeneity and tumor evolution and progression. By dividing tumors into high immune-ITH and low immune-ITH tumors based on immune-ITH scores, they characterized the composition, tumor evolution events and transcriptomic signatures associated with various degrees of immune-ITH and therefore concluded that intratumoural immune heterogeneity represents a hallmark of tumor evolution.

Unfortunately, the current analysis does not support the central claim "immune-ITH as a hallmark of tumor evolution and disease progression". The section about tumor evolution events is critically important to provide a basis for understanding the relationship between immune-ITH and tumor evolution. However, the analysis is only performed at an aggregated level, and analyzing the number of tumor evolution events is too superficial to gain any further insights beyond association. I believe diving into tumor evolution events to reveal certain possible patterns would greatly improve the manuscript.

The structure of the paper also seems not to be optimally organized as the section were scattered without a logical flow.

Major:

1. In the section of "Significant degree of immune-ITH in HCC", the immune landscape based on CyTOF is not well described or clearly presented. It is necessary to include key information such as the types of immune cells identified and the precise proportions of each immune cell type in each sample. I would suggest using reduced dimension techniques like t-SNE or UMAP to visualize the 38D data and a heatmap to show protein expression of each cell type.

2. Also, in the section "Significant degree of immune-ITH in HCC", makes claims such as the "relatively homogeneous distributions" of B016, the "marked degree of ITH" of H255 and "significant variations in the proportions" make no sense to me given that no quantitative analyses were conducted in this section. I would suggest rewriting this section, starting by introducing immune-ITH score and then describing the degree of immune-ITH based on this score.

3. The author found a relatively high correlation between immune-ITH and RNA-ITH. However, it is not clear how "RNA-ITH is indicative of tumor evolution" (line 221). The reference "The spatial organization of intra-tumour heterogeneity and evolutionary trajectories of metastases in hepatocellular carcinoma" used to support this claim has shown that the tumor evolutionary trajectories could be reconstructed using genomic profiles like somatic mutations and CNV but not RNA expression profile.

4. It is not clear that by what criteria did the author divide tumors into immune-ITH high group and low group? Would the cutoff selection impact the immune cell composition analysis (Fig. 3) and genomic events analysis (Fig 4.)?

5. The observation that high immune-ITH harbored a more immunosuppressive and exhausted TME is interesting and deserves further analysis to gain potential new insights. For example, is immune exhausted and suppressive TME associated with HBV infection status? If not, is it possible to delineate HBV-induced transcriptome signatures and HCC-induced transcriptome signatures? There is a lot of missed opportunities in this study.

6. At the patient level, it is not clear what the relationship is between immune-ITH and genomic

mutations. It is perhaps desirable to reconstruct phylogenetic relationship for each tumor using multi-region samples and then examine how immune-ITH developed along with phylogenetic tree.

Minor:

The resolution of all figures should be improved.

Reviewer #2 (Remarks to the Author): with expertise in HCC - tumor heterogeneity - immunogenomics

The here presented manuscript by Nguyen et al. aims to delineate heterogeneity in immune cell composition and its association with tumor evolution and cancer progression in hepatocellular carcinoma. The authors provide comprehensive data on immune cell contexture from different tumor regions using specimens from resections revealed by CyTOF data as well as WGS and bulk transcriptomic data.

The findings on an intratumoral heterogeneity in immune cell composition confirms already published papers describing the immune landscape in the context of intratumoral heterogeneity in HCC (Zhang et al, Gut, 2019). The key finding of the here presented study is an association of high immune ITH with a reduced immunoselective pressure and a more immunosuppressive microenvironment leading to tumor cell escape from immune-mediated cell death.

Although, the here presented findings are technically well performed, comprehensive and provide a conclusive overview of tumor-immune cell interactions, the novelty of the presented findings are overall limited in a very competitive field of research and context of recent studies on heterogeneity in immune cell composition in HCC (Zhang et al, Gut, 2019; Ma et al, Cancer Cell, 2019). In line with this analysis of neo-epitopes and influence on the adaptive immune system has recently also been demonstrated (Losic et al, Nat. comm., 2019). Furthermore, majority of findings remain largely descriptive and lack mechanistic explanation or functional validation as well as single cell approaches.

Major comments:

- It is not entirely clear if the presented analysis has been performed on one tumor region, combined samples from one tumor or different tumor regions from one tumor. Thus, conclusions on intratumoral heterogeneity as well as intertumoral heterogeneity are difficult to judge (e.g. Fig. 1c, 2, 3). Overall, majority of findings are only descriptive and lack a detailed mechanistic explanation. In particular, the link to tumor evolution is suggestive and lacks causality or functional validation.

- Several of the described findings overlap with results from recent reports. Common findings and key differences should be clearly defined and discussed:

- o Fig. 1 describes a heterogeneity in immune cell composition between patients as well as intratumoral regions using CyTOF data. This has been shown before using CyTOF data in Zheng et al., Gut, 2019 as well as on a single transcriptomic cell level in Ma et al, Cancer Cell, 2019.

- o CyTOF data (Fig. 1) should be extended and comprise key immune lineages. Myeloid cells, specifically MDSCs, have been shown to contribute to intratumoral immune cell reactions and should be explored. Furthermore, dendritic cells also have a major impact and should be investigated.

- o Fig. 2d describes the correlation of a transcriptomic heterogeneity and immune ITH based on CyTOF and bulk sequencing data. The influence of intratumoral heterogeneity on immune cells has recently been shown on a single cell level (Ma et al, Cancer cell, 2019). Similar, Fig. 4 describes HLA loss of heterozygosity and the association of immunoediting and immune ITH. Spatio-temporal heterogeneity and interactions between cancer and immune cells has been shown before in HCC (Losic et al, 2019, Nat. comm.). However, while regional neo-epitope and adaptive immune cell response as well as spatial clonal expansion of immune cells have been investigated, the HLA loss of heterozygosity is a new and interesting finding. Thus, a more comprehensive analysis of the immune cell composition including subgroup analysis would be interesting. This could be performed using CyTOF or additional TCR sequencing. Further, it would be interesting to define how loss of HLA-LOH affect T cell clonality or subgroup composition.

- Tumor specific transcriptome profiles were generated by subtracting genes contributing to immune subsets using CIBERSORT gene sets. Thus, genes associated with other cell populations such as stellate cells, hepatocytes, cancer associated fibroblasts are neglected. Single cell sequencing could be performed to conclusively define these profiles.

Minor comments:

- Recently, the results of IMbrave150 trial were reported in NEJM. This interesting study should be cited.
- Is Fig. 2a based on key immune lineages or subgroups?
- Fig. 3. Uncommon selection of exhaustion markers. The authors should at least provide two more common exhaustion markers such as LAG-3 or TIM3. Enrichment of exhausted T cells is only descriptive. In-vitro validation analyzing cytokine expression (TNF α , IFN γ) for functional analysis has been performed but only one exemplary dotplot is presented. Data from all patients or even regions should be shown as boxplots.
- Definition of immune pressure is not clear (line 250). The overall conclusion of the authors states that a high immune ITH is based on a low immune pressure. However, this statement needs to be clarified. A direct link to tumor evolution is lacking.
- Figure 5 is difficult to read.
- In general, utilized statistical tests should be mentioned and description should be extended.

Reviewer #3 (Remarks to the Author): with expertise in HCC - clinical

They tried to examine immune ITH correlating with tumor evolution and progression. They evaluated CyTOF, RNAseq and WGS data for the study. They have done so many studies and obtained so many correlation data combining multiple sampling from each tumor. However, 2 to 5-point sampling is not carefully executed to convince their sampling fully represent sub-clonal tumor heterogeneity or whole tumor heterogeneity of varying tumor size population (up to 15 cm tumor size). Historically tumor heterogeneity was evaluated based on histopathological features, while they rarely evaluated those features in their analyses. It is unique to separate low and high immune-ITH, however they fail to show its mechanistic background or clinic-pathological significance in detail.

Some specific comments are indicated below.

Introduction:

They should cite Hepatology 68: 1025-41 and Meylan et al, Clinical Cancer Research (online ahead of print), which describe immune ITH and HCC progression.

Results

CyTOF data had better been correlated with semi-quantitative IHC data in more detail to convince their low/high immune-ITH classification.

Mutation of CTNNB1 has been show to regulate immune microenvironments (Calderaro et al, J Hepatol 67: 727-38, Spranger et al, Nature 523: 231-5, Ruiz de Galarreta et al, Cancer Discovery 9: 1124-41) , which should be evaluated with low/high immune ITH.

Figure 4f, g may be statistically significant, but not so distinct to convince their interpretation.

Point-by-point response to reviewers' comments:

First of all, we would like to thank the reviewers for the valuable comments that have helped us to improve and revise our manuscript with the following major changes:

1. We added **new CyTOF analysis data** by Phenograph clustering (Levine et al., 2015) and our in-house CyTOF analytics pipeline (Yeo et al., 2020) (**New Fig 1b-f**) shown as **tSNE clusters (Fig.1b)** with the **heat map** illustration of the cluster **phenotypes (Fig.1c)** and the degree of **immune-ITH** according to abundance of these immune clusters (**Fig.1d** and **Supplementary Fig.2a**). We also showed that the immune-ITH scores according to tSNE clusters or manual gating of 15 immune subsets showed good concordance (**Fig. 1f**), demonstrating the **robustness in immune-ITH scoring**. To make space for these new data, we removed the original pie charts (Fig.1b) showing frequencies of immune lineages from only two representative cases.
2. More importantly, we added **an entirely new Fig. 2** to enhance the insights into **tumour evolutionary events** along immune-ITH: genomic **phylogenetic trees (Fig. 2a)**, copy-number variation (**CNV**) events (**Fig.2b** and **Fig.2c**) and identified **key mutation genes** either deleted or amplified in tumours with low vs high immune-ITH (**Fig.2b, bottom, Fig.2d** and **Supplementary Table 3**). This new data links immune-ITH to tumour evolutionary trajectory. The original Fig.2a-c showing immune-ITH quantifications has been shifted to Fig. 1 and original Fig.2d showing ITH correlations shifted to Supplementary Fig.6.

All the changes in the manuscript are marked by "underlined". Please note that due to the additional new supplementary data, the order of the original supplementary data was shifted accordingly.

Please find our point-by-point responses to the reviewers' comments below marked with "**Re**".

Reviewers' comments:

Reviewer #1 (Remarks to the Author): with expertise in immune landscape and HCC

In this paper, the author analyzed multi-omics data consisting of WGS, bulk RNA seq and, CyTOF from multi-region samples within the same HCC tumor from a 28 HCC patient cohort to reveal the relationship between intratumoural immune heterogeneity and tumor evolution and progression. By dividing tumors into high immune-ITH and low immune-ITH tumors based on immune-ITH scores, they characterized the composition, tumor evolution events and transcriptomic signatures associated with various degrees of immune-ITH and therefore concluded that intratumoural immune heterogeneity represents a hallmark of tumor evolution.

Unfortunately, the current analysis does not support the central claim "immune-ITH as a hallmark of tumor evolution and disease progression". The section about tumor evolution events is critically important to provide a basis for understanding the relationship between immune-ITH and tumor evolution. However, the analysis is only performed at an aggregated level, and analyzing the number of tumor evolution

events is too superficial to gain any further insights beyond association. I believe diving into tumor evolution events to reveal certain possible patterns would greatly improve the manuscript.

The structure of the paper also seems not to be optimally organized as the section were scattered without a logical flow.

Re: We agree to this assessment and in fact we already have more genomic data that could support this claim. We have now added a **new figure 2** exploring the tumour evolution trajectory along with immune-ITH. We have also improved on the logical flow of the manuscript. Please see the details below in point-by-point response to each comment.

Major:

1. In the section of “Significant degree of immune-ITH in HCC”, the immune landscape based on CyTOF is not well described or clearly presented. It is necessary to include key information such as the types of immune cells identified and the precise proportions of each immune cell type in each sample. I would suggest using reduced dimension techniques like t-SNE or UMAP to visualize the 38D data and a heatmap to show protein expression of each cell type.

Re: We have now added the **t-SNE** representation of our CyTOF data by Phenograph clustering (Levine et al., 2015) and our in-house EPIC CyTOF analytics pipeline (Yeo et al., 2020) (new **Fig.1b**), the **heat map** illustration of the cluster **phenotypes** (new **Fig.1c**) and the degree of **immune-ITH** according to abundance of these immune clusters (new **Fig.1d** and new **Supplementary Fig.2a**). Most importantly, we showed a strong concordance between immune-ITH scores according to tSNE clusters and manual gating of 15 immune subsets (new **Fig. 1f**), demonstrating the **robustness in immune-ITH scoring**.

Shown below for your kind reference, the **tSNE clusters (Fig.1b)** and **heat map** for the protein markers expression or the phenotypes of these clusters (**Fig.1c**).

2. Also, in the section “Significant degree of immune-ITH in HCC”, makes claims such as the “relatively homogeneous distributions” of B016, the “marked degree of ITH” of H255 and “significant variations in the proportions” make no sense to me given that no quantitative analyses were conducted in this section. I would suggest

rewriting this section, starting by introducing immune-ITH score and then describing the degree of immune-ITH based on this score.

Re: This is indeed a good suggestion. We actually did provide quantification of ITH on immune cell density later in the original Fig.2c as the **standard division of immune cell densities between at least 10 different tissue regions of size 3mm²**. We have now moved this quantification data to **Fig.1h** to better show its correlation with immune-ITH quantified from CyTOF data.

3. The author found a relatively high correlation between immune-ITH and RNA-ITH. However, it is not clear how “RNA-ITH is indicative of tumor evolution” (line 221). The reference “The spatial organization of intra-tumour heterogeneity and evolutionary trajectories of metastases in hepatocellular carcinoma” used to support this claim has shown that the tumor evolutionary trajectories could be reconstructed using genomic profiles like somatic mutations and CNV but not RNA expression profile.

Re: Indeed, we agree and have now included the **CNV** events and the **specific mutations** which correlated with immune-ITH in the **new Fig. 2b-c** (also shown below for your kind reference). From the new Fig. 2, we demonstrated higher events of CNVs, indicating higher genomic instability in tumours with high immune-ITH (**Fig.2b,c**) and identified key mutation genes either deleted or amplified (**Fig.2b, bottom, Fig.2d** and **Supplementary Table 3**). This new data now further links immune-ITH to tumour evolutionary trajectory. Please note the paragraph describing the new Fig. 2 was added with “Underlined” in the revised manuscript.

4. It is not clear that by what criteria did the author divide tumors into immune-ITH high group and low group? Would the cutoff selection impact the immune cell composition analysis (Fig. 3) and genomic events analysis (Fig 4.)?

Re: We actually explained the stratification in Supplementary Fig.5a, where we divided the tumours into immune-ITH high or low groups based on **median level of immune-ITH scores**. Hence for both Fig. 3 and Fig. 4 and grouping were based this immune-ITH grouping. We have now added this explanation "underlined" before the analysis leading to Fig. 3 and 4 for further clarification.

5. The observation that high immune-ITH harbored a more immunosuppressive and exhaustive TME is interesting and deserves further analysis to gain potential new insights. For example, is immune exhausted and suppressive TME associated with HBV infection status? If not, is it possible to delineate HBV-induced transcriptome signatures and HCC-induced transcriptome signatures? There is a lot of missed opportunities in this study.

Re: We did actually examine HBV in **supplementary Fig.10b**, that immune-ITH was not linked to hepatitis viral infection. We believe this point may have been missed and hence we added a sentence (underlined) in the text to highlight this point again. We believe this data indicates that the immune-ITH is a general immune evolutionary event in HCC that is not dependent on viral hepatitis.

6. At the patient level, it is not clear what the relationship is between immune-ITH and genomic mutations. It is perhaps desirable to reconstruct phylogenetic relationship for each tumor using multi-region samples and then examine how immune-ITH developed along with phylogenetic tree.

Re: Indeed, this is a very good suggestion. We have now added the **phylogenetic trees showing both RNA- or DNA-ITH** in either low or high immune-ITH tumours (**new Fig. 2a**). Overall, we observed concordance between immune-ITH and RNA- or DNA-ITH where tumours with low immune-ITH showed shorter RNA or DNA branch distances between tumour sectors compared to those with high immune-ITH (**new Fig.2a** and also shown below). For each patient, the mutations are different but the genes showing overall mutations comparing immune-ITH high or low are shown in **new Fig. 2b and 2d**.

Minor:

The resolution of all figures should be improved.

Re: We apologize for the poor resolution when converted to PDF files and have now submitted the JPEG files instead.

Reviewer #2 (Remarks to the Author): with expertise in HCC - tumor heterogeneity - immunogenomics

The here presented manuscript by Nguyen et al. aims to delineate heterogeneity in immune cell composition and its association with tumor evolution and cancer progression in hepatocellular carcinoma. The authors provide comprehensive data on immune cell contexture from different tumor regions using specimens from resections revealed by CyTOF data as well as WGS and bulk transcriptomic data. The findings on an intratumoral heterogeneity in immune cell composition confirms already published papers describing the immune landscape in the context of intratumoral heterogeneity in HCC (Zhang et al, Gut, 2019). The key finding of the here presented study is an association of high immune ITH with a reduced immunoselective pressure and a more immunosuppressive microenvironment leading to tumor cell escape from immune-mediated cell death.

Although, the here presented findings are technically well performed, comprehensive and provide a conclusive overview of tumor-immune cell interactions, the novelty of the presented findings are overall limited in a very competitive field of research and context of recent studies on heterogeneity in immune cell composition in HCC (Zhang et al, Gut, 2019; Ma et al, Cancer Cell, 2019). In line with this analysis of neo-epitopes and influence on the adaptive immune system has recently also been demonstrated (Losic et all, Nat. comm., 2019). Furthermore, majority of findings remain largely descriptive and lack mechanistic explanation or functional validation as well as single cell approaches.

Re: Indeed, this is a highly competitive field with several previous publications however our findings offer new insights and novel findings on the relevance of immune-ITH as hallmark of tumour evolution, which is largely distinct from these previous studies (please see the detail point-by-point response below). Also, we would like to stress that **CyTOF** too is a **single-cell proteomics approach**. Even though we do have scRNA seq data on a subset of cohort (n=9 patients), this dataset is separately analysed for myeloid cells remodelling and have just been accepted in Cell (*Sharma et al. Cell. Accepted*), hence we did not include here. Further comments on mechanistic insights and functional validation were provided below.

Major comments:

- It is not entirely clear if the presented analysis has been performed on one tumor region, combined samples from one tumor or different tumor regions from one tumor. Thus, conclusions on intratumoral heterogeneity as well as intertumoral heterogeneity are difficult to judge (e.g. Fig. 1c, 2, 3). Overall, majority of findings are only descriptive and lack a detailed mechanistic explanation. In particular, the link to tumor evolution is suggestive and lacks causality or functional validation.

Re: The data is actually generated per tumour region from multiple regions of each tumour therefore the conclusion on “**intratumoural**” immune heterogeneity. To enhance the clarity, we added “We employed Phenograph clustering(Levine et al., 2015) and our in-house CyTOF analytics pipeline(Yeo et al., 2020) on data generated from all tumour sectors” in the result section.

To address the comment on link to tumour evolution, we have now added new data exploring additional tumour evolutionary events along immune-ITH (**new data Fig.2**) including genomic **phylogenetic trees** (**Fig. 2a**), copy-number variation (**CNV**) (**Fig.2b** and **Fig.2c**) and identification of **key mutation genes** either deleted or amplified (**Fig.2b, bottom, Fig.2d** and **Supplementary Table 3**) along immune-ITH. This new data enhances link of immune-ITH to tumour evolutionary trajectory.

For functional aspects, we provided the relationship between immune-ITH and immune pressure (**Fig.3**, including a **functional study with *in-vitro* immune cell activation in Fig.3d**) which potentially drives these tumour evolutionary events: CNV (**new Fig.2**), mutational and neoantigens burden (**Fig. 4a-4b**) and immunoediting (**Fig.4c-4f**). We further offered mechanistic insights from the molecular signature and transcriptome-immune cell networks according to immune-ITH levels (**Fig. 5**). Therefore, we believe we have provided substantial evidence to support immune-ITH as a hallmark of tumour evolution.

- Several of the described findings overlap with results from recent reports. Common findings and key differences should be clearly defined and discussed:

o Fig. 1 describes a heterogeneity in immune cell composition between patients as well as intratumoral regions using CyTOF data. This has been shown before using CyTOF data in Zheng et al., Gut, 2019 as well as on a single transcriptomic cell level in Ma et al, Cancer Cell, 2019.

Re: We actually did cite these two papers in the introduction section. However, we now provided a clearer explanation to highlight the novelty that our current study offered against these previous studies (changes underlined in introduction). **Zhang et al Gut 2019 (ref 12)** used multiomic analysis to reveal comprehensive ITH in HCC but do not draw any conclusion on the biological or clinical relevance of the degree of immune-ITH. **Ma et al Cancer Cell 2019 (ref 6)** on the other hand described tumour cell transcriptomic diversity as linked to response to check-point blockage (ICB). They too did not actually explore immune-ITH and its biological or clinical significance as offered in our current study.

o CyTOF data (Fig. 1) should be extended and comprise key immune lineages. Myeloid cells, specifically MDSCs, have been shown to contribute to intratumoral immune cell reactions and should be explored. Furthermore, dendritic cells also have a major impact and should be investigated.

Re: In the new Fig. 1b-f, we have comprehensively covered all possible TILs including the myeloid subsets with tSNE presentation of all our TILs CyTOF data after unbiased down-dimension and Phenograph clustering (Levine et al., 2015) using our in-house EPIC CyTOF analytics pipeline (Yeo et al., 2020) (**Fig. 1b, 1c and 1d and supplementary Fig. 2**). Also the original manual gating validation of our CyTOF data actually did encompass all TILs including CD14+ myeloid subsets (**Fig.1e and Supplementary Fig.3**). Furthermore, we showed good concordance between the immune-ITH scores according to tSNE clusters or manual gating of 15 immune subsets (**Fig. 1f**), demonstrating the **robustness in immune-ITH scoring**. Therefore, we did consider all TILs including all the myeloid subsets in the immune-ITH calculation.

o Fig. 2d describes the correlation of a transcriptomic heterogeneity and immune ITH based on CyTOF and bulk sequencing data. The influence of intratumoral heterogeneity on immune cells has recently been shown on a single cell level (Ma et al, Cancer cell, 2019). Similar, Fig. 4 describes HLA loss of heterozygosity and the association of immunoediting and immune ITH. Spatio-temporal heterogeneity and interactions between cancer and immune cells has been shown before in HCC (Losic et al, 2019, Nat. comm.). However, while regional neo-epitope and adaptive immune cell response as well as spatial clonal expansion of immune cells have been investigated, the HLA loss of heterozygosity is a new and interesting finding. Thus, a more comprehensive analysis of the immune cell composition including subgroup analysis would be interesting. This could be performed using CyTOF or additional TCR sequencing. Further, it would be interesting to define how loss of HLA-LOH affect T cell clonality or subgroup composition.

Re: Firstly, **Ma et al. (ref 6)** employed scRNA-seq on multiple tumour sectors and concluded that tumour cell transcriptomic diversity is linked to response to ICB, rather than immune-ITH as we measured in our current study based on CyTOF analysis on isolated TILs. We actually cited this in the first paragraph of the introduction now "underlined" for your kind reference.

Secondly, in contrast to **Losic et al 2019, Nat Comm (ref 14)** (which we also cited in our manuscript "underlined"), focused mainly on how TCR clonal heterogeneity is correlated to tumour genomic ITH, while our study focused on immune-ITH based on the composition of TILs as an immune evolutionary events.

We unfortunately do not have the full TCR clonality data due to the insufficient depth in our bulk RNA seq data. Even though we do have the scRNA seq TCR data but they are only available for a subset of the cohort. Furthermore, as *Losic et al.* had already explored the angle of TCR clonality ITH, we would therefore herein focus on immune-ITH of TILs based on the comprehensive immune cell analysis using CyTOF (single-cell immunomics technology) and immunohistochemistry (immune cell densities analysis).

- Tumor specific transcriptome profiles were generated by subtracting genes contributing to immune subsets using CIBERSORT gene sets. Thus, genes associated with other cell populations such as stellate cells, hepatocytes, cancer associated fibroblasts are neglected. Single cell sequencing could be performed to conclusively define these profiles.

Re: The key reason why we subtract the genes from immune subsets using CIBERSORT is to remove genes contributed by the immune cells in order to study their interaction with other cells within the tumour. Therefore, these other stromal subsets are actually not neglected. Also, because our focus is on the immune subsets interaction, we did not go into each of these other subsets as it will be over and beyond the current scope.

Minor comments:

- Recently, the results of IMbrave150 trial were reported in NEJM. This interesting study should be cited.

Re: At the time of submission it was not yet published and now we have cited this NEJM paper (**ref 5**) whenever IMBrave150 trial is mentioned.

- Is Fig. 2a based on key immune lineages or subgroups?

Re: The original Fig.2a immune-ITH quantification was based on **all key immune subsets in TILs** not immune lineages. It is now moved to the **bottom of revised Fig.1e** (see also attached below) to improve clarity of the message.

- Fig. 3. Uncommon selection of exhaustion markers. The authors should at least provide two more common exhaustion markers such as LAG-3 or TIM3. Enrichment of exhausted T cells is only descriptive. In-vitro validation analyzing cytokine expression (TNF α , IFN γ) for functional analysis has been performed but only one exemplary dotplot is presented. Data from all patients or even regions should be shown as boxplots.

Re: Thank u for the suggestion. To expand our scope to explore other immune exhaustion markers, we have now added the new data on Tim3+/- and Lag3+/- CD8+ T cell in **revised Fig. 3a** and **revised supplementary Fig.8a** (also shown below for your kind reference). Tim-3+CD8+ T cells, which are more abundant than Lag-3+CD8+ T cells showed consistent distributions with PD-1+/-CD8+ T cells, whereby exhausted Tim-3+ or PD-1+CD8+ T cells are enriched in high immune-ITH tumours (**revised Fig.3a**). Lag-3+CD8+ T cells are rather small populations (~1-2%) compared to the dominant PD-1+ or Tim-3+ T cells subsets and do not show any significant differences in low vs high immune-ITH tumours (**revised supplementary Fig.8a**). This additional data further supports a more immunosuppressive TME in tumours with high immune-ITH.

In fact, all cytokines data (per region) were actually presented in **Fig. 3d** even though 3c shows only a representative dot plot. To enhance clarity, we relabel the title of this graph as “TNF α ⁺IFN γ ⁺ T cell” (**revised Fig.3d** and below).

- Definition of immune pressure is not clear (line 250). The overall conclusion of the authors states that a high immune ITH is based on a low immune pressure. However, this statement needs to be clarified. A direct link to tumor evolution is lacking.

Re: To clarify this, as it has been known that genomic-ITH is an established hallmark of tumour evolution, the current study aims to explore if **immune-ITH** could be a novel hallmark of tumour evolution. With this aim in mind, we explored and showed that increased immune-ITH is linked to reduced immune pressure, indicating immune evolution towards immune exhaustion and suppression (**Fig.3a-d**). Its direct relationship with multiple tumour evolution events were also explored in **new Fig. 2 (CNV and tumour genomic-ITH)** and **Fig. 4 (immunoediting events)**. All of these tumour evolutionary events are directly linked to low or high immune-ITH. Therefore, we conclude that immune-ITH as one of the key events or manifestation of tumour evolution linking all these events together that leads to tumour progression. This is the key novelty and centre focus of our current manuscript.

- Figure 5 is difficult to read.

Re: The data is indeed complex. To improve its clarity, we **have reduced complexity in Fig. 5a and 5b** by removing the pathways and genes labelled on the gene vs immune cells networks. In fact, the pathways involved in the tumour transcriptome vs immune cells interaction networks can be referred to in **Fig. 5c and 5d**. The key discovery here is on the molecular pathways involved in immune-ITH low vs high tumours. We found metabolism pathways to be involved in maintaining

immune status in tumours with low immune-ITH and pathways related to tumour proliferation and progression as enriched in tumours with high immune-ITH.

- In general, utilized statistical tests should be mentioned and description should be extended.

Re: We did mention each test used in figure legends which might be missed. We have now highlighted these again as "underlined" in each figure legend for your kind reference.

Reviewer #3 (Remarks to the Author): with expertise in HCC - clinical

They tried to examine immune ITH correlating with tumor evolution and progression. They evaluated CyTOF, RNAseq and WGS data for the study. They have done so many studies and obtained so many correlation data combining multiple sampling from each tumor.

However, 2 to 5-point sampling is not carefully executed to convince their sampling fully represent sub-clonal tumor heterogeneity or whole tumor heterogeneity of varying tumor size population (up to 15 cm tumor size). Historically tumor heterogeneity was evaluated based on histopathological features, while they rarely evaluated those features in their analyses. It is unique to separate low and high immune-ITH, however they fail to show its mechanistic background or clinic-pathological significance in detail.

Re: Thank u for the thorough evaluation of our study. First of all, the tumour sector sampling was done following strict protocol covering across the diameter of epicentre of each tumour with a clearly defined gap between each sector as illustrated in **Supplementary Fig.1**. In fact, we did also examine the histological features as much as we can to support our claims (**Fig. 1h and Fig. 3b**). We offered the mechanistic insights of immune-ITH on immune pressure (**Fig.3**) and tumour evolutionary events in **new Fig. 2 (CNV and tumour genomic-ITH)** and **Fig. 4 (immunoediting events)**. Last but not least, we provided the clinical relevance of immune-ITH where it predicts outcome of HCC patients, which was also confirmed in additional two public cohorts (**Fig.6**). Hence we believe we have provided substantial evidence to support immune-ITH as a hallmark of tumour evolution.

Some specific comments are indicated below.

Introduction:

They should cite Hepatology 68: 1025-41 and Meylan et al, Clinical Cancer Research (online ahead of print), which describe immune ITH and HCC progression.

Re: We have now cited these two histology papers in introduction marked as "underlined" (**Meylan et al. as ref 10** and **Kurebayashi et al as ref 11**). However, they are distinct from our current study as they described the immunological features of HCC by IHC in relationship to clinical outcome and did not explore on immune-ITH as tumour evolutionary event as we did. Specifically, **Kurebayashi et al** analyse the immune landscape on multiple regions of HCC but segregated the tumours to immune hi, mid or low and correlated with clinical findings while **Meylan et al** explored the impact of tertiary lymphoid structures (TLS) in early hepatic lesion.

Results

CyTOF data had better been correlated with semi-quantitative IHC data in more detail to convince their low/high immune-ITH classification.

Re: We have in fact profiled and quantified 10 regions per tumour each of size 3mm² for the IHC data and calculated their standard deviation as indicator of immune cell densities ITH. We moved this data to Fig.1 to better correlate with our CyTOF immune-ITH data (**revised Fig. 1h** and also shown below). We have also underlined the text describing this data for your kind reference.

Mutation of CTNNB1 has been show to regulate immune microenvironments (Calderaro et al, J Hepatol 67: 727-38, Spranger et al, Nature 523: 231-5, Ruiz de Galarreta et al, Cancer Discovery 9: 1124-41), which should be evaluated with low/high immune ITH.

Re: This is indeed a good suggestion. We have explored the mutation of *CTNNB1*, however immune-ITH is not related to *CTNNB1* mutation (**new Fig.2d**) nor immune-exclusion (**Supplementary Fig7a and 7b**) (also shown below for your reference). This also means that the ITH data is not related to high or low immune infiltration, rather it is a phenomenon on homogeneity or heterogeneity in distribution of these immune cells. This is in fact the key novelty and main focus of our current discovery. In fact, this interesting discovery showing immune phenotypic changes and immune exclusion as two independent tumour escape events along tumour progression is now being explored in our new manuscript under preparation. It extends beyond our current findings on the significance of immune-ITH as hallmark of tumour evolution and examined in depth of the evolutionary trajectory of immune landscapes during tumour progression.

New Fig. 2d: immune-ITH vs CTNNB1 mutation

New Supplementary Fig. 7: Immune-ITH versus immune exclusion

Figure 4f, g may be statistically significant, but not so distinct to convince their interpretation.

Re: We must admit that the current cohort of patient is rather small and furthermore due to patient-to-patient diversity hence the low statistical significance is rather expected. However, the main findings were measured using few key independent technologies: CyTOF and IHC analyses (**Fig.1**), the tumour evolutionary events explored across different angles: CNV (**new Fig.2**), mutational and neoantigens burden (**Fig. 4a-4b**) and immunoediting (**Fig.4c-4f**). Furthermore, we also validated our findings in two independent cohorts of HCC patients which demonstrated the robustness of our data (**Fig. 6f**). We would therefore like to conclude and hopefully convince the reviewer of the robustness and credibility of our findings and conclusion.

References:

Levine, J.H., Simonds, E.F., Bendall, S.C., Davis, K.L., Amir el, A.D., Tadmor, M.D., Litvin, O., Fienberg, H.G., Jager, A., Zunder, E.R., *et al.* (2015). Data-Driven Phenotypic Dissection of AML Reveals Progenitor-like Cells that Correlate with Prognosis. *Cell* 162, 184-197.

Yeo, J.G., Wasser, M., Kumar, P., Pan, L., Poh, S.L., Ally, F., Arkachaisri, T., Lim, A.J.M., Leong, J.Y., Lai, L., *et al.* (2020). The Extended Polydimensional Immunome Characterization (EPIC) web-based reference and discovery tool for cytometry data. *Nat Biotechnol.*

Reviewer #1 (Remarks to the Author):

In general, the revised manuscript is better organized. The newly added CyTOF data analysis and tumor evolutionary events analysis significantly improved the logic of the whole manuscript and strengthened the main claim of this manuscript. I think the authors have addressed most of my concerns to a sufficient degree and also provided several suggestive evidences to link immune-ITH and tumor evolution and progression.

In my view, one concern remains regarding the data and interpretation of the CyTOF that should be addressed.

1) The CyTOF analysis generated 30 clusters, but the heatmap (Fig 1c) only comprised several main immune lineages. Although the author showed concordance of quantifying the degrees of immune-ITH using two different approaches, I think it is necessary to present whether the CyTOF based clusters is consistent with the Manual gating clusters. It is not clear if the CyTOF clusters would reveal one or several novel immune cell types.

2) Following 1), based on the cell types identified by CyTOF, it is not clear which immune subsets are enriched in the low and high immune-ITH tumors. I believe an integrated analysis of CyTOF and manual gating would strengthen the claim "Immune exhaustive and suppressive TME in tumours with high immune-ITH".

Reviewer #2 (Remarks to the Author):

This is the revised version of the manuscript.

The authors carefully and experimentally addressed most of the raised concerns. In particular, the missing link to tumor evolution has comprehensively been delineated and described in detail. Further, figures and figure legends have been significantly improved. The revision strengthened the manuscript. Overall, the study is well performed and conclusive. However, concerns related to the limited novelty remain. Major parts of the here addressed topic have been described in previous studies including Ma et al. (Cancer Cell) and Losic et al (Nature communications).

Reviewer #3 (Remarks to the Author):

The reviewer is still not sure how immune ITH correlate histopathological heterogeneity and histopathological tumor evolution. It is well known that tumor differentiation and nodule-in-nodule appearance represent tumor evolution in HCC. Their strict tumor sampling protocol simply based on anatomy, not histopathology. New ref 11 (Hepatology 68) already mention immune features based on histopathological transitional changes.

Revised Fig 1h is difficult to recognize homogeneous/heterogeneous distribution for the reviewer.

Fig. 2d "Amplification of CTNNB1": Is it correct, not mutation?

Point-by-point response to reviewers' comments:

We are grateful for the reviewers' comments which help improve our manuscript tremendously. We aim to address their remaining concerns with the point-by-point response below marked with **Re**.

Reviewer #1 (Remarks to the Author):

In general, the revised manuscript is better organized. The newly added CyTOF data analysis and tumor evolutionary events analysis significantly improved the logic of the whole manuscript and strengthened the main claim of this manuscript. I think the authors have addressed most of my concerns to a sufficient degree and also provided several suggestive evidences to link immune-ITH and tumor evolution and progression.

In my view, one concern remains regarding the data and interpretation of the CyTOF that should be addressed.

1) The CyTOF analysis generated 30 clusters, but the heatmap (Fig 1c) only comprised several main immune lineages. Although the author showed concordance of quantifying the degrees of immune-ITH using two different approaches, I think it is necessary to present whether the CyTOF based clusters is consistent with the Manual gating clusters. It is not clear if the CyTOF clusters would reveal one or several novel immune cell types.

Re: Indeed, CyTOF analysis generated 30 clusters which is more than the 15 immune clusters we gated manually. But there were two points to note:

1. Some of these clusters from CyTOF have similar phenotypes which often resulted in small subsets that are very hard to gate manually and could potentially be combined into bigger and more biologically relevant subsets as we did for the 15 manually gated subsets. In fact, it is recommended to check the data using manual gating which is more accurate and unbiased against any outliers.
2. Our manual gating has already taken into considerations of all possible CD45+ immune subsets (Supplementary Fig. 3a) hence we did not miss any potentially important subsets.

2) Following 1), based on the cell types identified by CyTOF, it is not clear which immune subsets are enriched in the low and high immune-ITH tumors. I believe an integrated analysis of CyTOF and manual gating would strengthen the claim "Immune exhaustive and suppressive TME in tumours with high immune-ITH".

Re: To address this comment, we did compare all 30 CyTOF clusters from tumours with high versus low immune-ITH using two-tailed unpaired Mann-whitney U test. We found CyTOF clusters with phenotypes **consistent** to our manually gated subsets as enriched in high versus low immune-ITH (see data below). We found clusters 4 (Treg), 7 (PD-1+ exhausted CD8+ T cells), 0 (Gzmb⁻ memory CD4 T cells) were enriched in tumours with high immune-ITH as consistent with our manually gated data in Fig. 3a. Conversely, we found enriched in tumours with low immune-ITH clusters 16 & 2 (CD69+ and CD69- NK cells), 22 & 3 (PD-1-Gzmb+ active CD8+ T

cells) and 23 (GzmB+ memory CD4+ T cells), again consistent with our manually gated data in Fig. 3a. The only cluster we found that was not validated by manual gating is the NKT cluster 15 (Supplementary Fig. 8a), of which we believe the significance could potentially be driven by outliers given that it is a relatively small subset (3-8% of total immune cells).

This result once again demonstrates the **robustness** of our data. Due to the final stage of manuscript revision and the confirmative but yet not additionally informative nature of this data, we propose **not to include** this data to the revised manuscript.

Reviewer #2 (Remarks to the Author):

This is the revised version of the manuscript. The authors carefully and experimentally addressed most of the raised concerns. In particular, the missing link to tumor evolution has comprehensively been delineated and described in detail. Further, figures and figure legends have been significantly improved. The revision strenghtned the manuscript. Overall, the study is well performed and conclusive. However, concerns related to the limited novelty remain. Major parts of the here addressed topic have been described in previous studies including Ma et al. (Cancer Cell) and Losic et al (Nature communications).

Re: We would again emphasize on the differences between our study with these two previous studies cited herein (this information was in fact provided in the previous point-by-point response to reviewers’ comments during the first revision):

Ref 6 **Ma et al** (Cancer Cell) employed scRNA-seq to analyze the multiple tumour sectors before immune checkpoint blockade (ICB) and concluded that tumour cell transcriptomic diversity is linked to response. They did not explore the biological and clinical relevance of immune-ITH as we did in our current study.

Ref 14 **Losic et al** (Nature communications) focused mainly on how TCR clonal heterogeneity is correlated to tumour genomic ITH, while our study focused on immune-ITH based on the composition of TILs as an immune evolutionary events that is relevant to clinical outcome.

We did in fact cite both of these articles (ref 6 and 14) with brief statements to highlight our novelty beyond their findings in the introduction.

Reviewer #3 (Remarks to the Author):

The reviewer is still not sure how immune ITH correlate histopathological heterogeneity and histopathological tumor evolution. It is well known that tumor differentiation and nodule-in-nodule appearance represent tumor evolution in HCC. Their strict tumor sampling protocol simply based on anatomy, not histopathology. New ref 11 (Hepatology 68) already mention immune features based on histopathological transitional changes.

Re: We agree that histopathological heterogeneity is linked to tumour evolution in HCC, however the current findings offered a novel discovery on the heterogeneity of the immune landscapes as hallmark of tumour evolution, which was not previously described. In fact, we did show that fibrosis is linked to enhanced immune-ITH (Fig. 6a), indicating its association to histopathological features of the tumour. We believe further exploration of the link of immune-ITH with histopathological features will require another independent study beyond the scope of this current manuscript, which focused primarily on immune-ITH and its relationship with genomic evolution and clinical outcome of HCC.

In addition, our tumour sampling protocol actually took representative samples across the centre diameter of the tumours (Supplementary Fig.1a), which is designed to represent the entire tumours from margin to core as unbiased as possible.

Lastly, despite the fact that Ref 11 by **Kurebayashi et al** (Hepatology 2018) analysed the immune landscape on multiple regions of HCC, they segregated the tumours to immune hi, mid or low to correlate with clinical findings. It is in fact well known that highly immune-infiltrated tumours, particularly if infiltrated by CD8 T cells (as also shown by our own study in **Chew et al Gut 2012**), is linked to superior clinical outcome. Kurebayashi et al however did not demonstrate the relevance of immune heterogeneity as an evolutionary events of tumours as we have shown herein.

Revised Fig 1h is difficult to recognize homogeneous/heterogenous distribution for the reviewer.

Re: Revised Fig. 1h was actually the original Fig. 1d but slightly minimized to make space for the correlation graphs on the right, which were added during revision. We have now done minor edits by reducing the size of the legend and keep it to just the image on the right to improve on image clarity. We can see that in the image on the left the T cells distribution was dispersed across the tumour (homogeneous) while image on the right showed concentrated spots of T cells distribution (heterogenous). Furthermore this IHC heterogeneity is well correlated with ITH detected by CyTOF (Fig.1h, right).

Fig. 2d “Amplification of CTNNB1”: Is it correct, not mutation?

Re: Yes, amplification or deletion of the CTNNB1 genes are the correct terms to use here as these are not point mutation or INDELS but rather amplification or deletion of the copy number of CTNNB1 gene. We have revised the text in the result from “mutation” to “CNV” (copy-number variation).